# Constrained Adaptive Attack: Effective Adversarial Attack Against Deep Neural Networks for Tabular Data

**Thibault Simonetto**
University of Luxembourg
Luxembourg
`thibault.simonetto@uni.lu`

**Salah Ghamizi**
LIST / RIKEN AIP
Luxembourg
`salah.ghamizi@list.lu`

**Maxime Cordy**
University of Luxembourg
Luxembourg
`maxime.cordy@uni.lu`

## Abstract

State-of-the-art deep learning models for tabular data have recently achieved acceptable performance to be deployed in industrial settings. However, the robustness of these models remains scarcely explored. Contrary to computer vision, there are no effective attacks to properly evaluate the adversarial robustness of deep tabular models due to intrinsic properties of tabular data, such as categorical features, immutability, and feature relationship constraints. To fill this gap, we first propose CAPGD, a gradient attack that overcomes the failures of existing gradient attacks with adaptive mechanisms. This new attack does not require parameter tuning and further degrades the accuracy, up to 81% points compared to the previous gradient attacks. Second, we design CAA, an efficient evasion attack that combines our CAPGD attack and MOEVA, the best search-based attack. We demonstrate the effectiveness of our attacks on five architectures and four critical use cases. Our empirical study demonstrates that CAA outperforms all existing attacks in 17 over the 20 settings, and leads to a drop in the accuracy by up to 96.1% points and 21.9% points compared to CAPGD and MOEVA respectively while being up to five times faster than MOEVA. Given the effectiveness and efficiency of our new attacks, we argue that they should become the minimal test for any new defense or robust architectures in tabular machine learning.

## 1 Introduction

Evasion attack is the process of slightly altering an original input into an *adversarial example* designed to force a machine learning (ML) model to output a wrong decision. Robustness to adversarial examples is a problem of growing concern among the secure ML community, with over 10,000 publications on the subject since 2014 [7]. Recent studies also report real-world occurrences of evasion attacks, which demonstrate the importance of studying and defending against this phenomenon [20].

While research has studied the robustness of deep learning models in Computer Vision (CV) and Natural Language Processing (NLP) tasks, many real-world applications instead deal with tabular data, including in critical fields like finance, energy, and healthcare. If classical "shallow" models (e.g. random forests) have been the go-to solution to learn from tabular data [21], deep learning models are becoming competitive [5]. This raises anew the need to study the robustness of these models.

However, robustness assessment for tabular deep learning models brings a number of new challenges that previous solutions — because they were originally designed for CV or NLP tasks — do not consider. One such challenge is the fact that tabular data exhibit *feature constraints*, i.e. complex relationships and constraints across features. The satisfaction of these feature constraints can be a

non-convex or even non-differentiable problem; this implies that established evasion attack algorithms relying on gradient computation do not create valid adversarial examples (i.e., constraint satisfying) [18]. Meanwhile, attacks designed for tabular data also ignore feature type constraints [3] or, in the best case, consider categorical features without feature relationships [40, 41, 4] and are evaluated on datasets that exclusively contain such features. This restricts their application to other domains that present heterogeneous feature types.

The only published evasion attacks that support feature constraints are *Constrained Projected Gradient Descent* (CPGD) and *Multi-Objective Evolutionary Adversarial Attack* (MOEVA) [35]. CPGD is an extension of the classical gradient-based PGD attack with a new loss function that encodes how far the generated examples are from satisfying the constraints. Although theoretically elegant and practically efficient, this attack suffers from a low success rate due to its difficulty to converge toward both model classification and constraint satisfaction [35]. Conversely, MOEVA is based on genetic algorithms. It offers an outstanding success rate compared to CPGD and works on shallow and deep learning models. However, it is computationally expensive and requires numerous hyper-parameters to be tuned (population size, mutation rate, generations, etc.). This prevents this attack from scaling to larger models and datasets.

Overall, research on adversarial robustness for tabular machine learning in general (and tabular deep learning in particular) is still in its infancy. This is in stark contrast to the abundant literature on adversarial robustness in CV [28] and NLP tasks [15]. Given this limited state of knowledge, the **objective** of this paper is to propose novel and effective attack methods for tabular models subject to feature constraints.

We hypothesize that gradient-based algorithms have not been explored adequately in [35] and that the introduction of dedicated adaptive mechanisms can outperform CPGD. To verify this, we design a new adaptive attack, named *Constrained Adaptive PGD* (CAPGD), whose only free parameter is the number of iterations and that does not require additional parameter tuning (Section 4). We demonstrate that the different mechanisms we introduced in CAGPD contribute to improving the success rate of this attack compared to CPGD, by 81% points. Across all our datasets, the set of adversarial examples that CAPGD generates subsumes all the examples generated by any other gradient-based method. Furthermore, CAPGD is 75 times faster than MOEVA, while the latter reaches the highest success rate across all datasets.

These results motivate us to design *Constrained Adaptive Attack* (CAA), an adaptive attack that combines our new gradient-based attack (CAPGD) with MOEVA for an increased success rate at a lower computational cost. Our experiments show that CAA reaches the highest success rate for all models/datasets we considered, except in one case where CAA is second-best. With this attack, we offer a strong baseline for future research on evasion attacks for tabular models, which should become the minimal test for robust tabular architectures and other defense mechanisms.

**Our contributions can be summarized as follows:**

1. We design a new parameter-free attack, CAPGD that introduces momentum and adaptive steps to effectively evade tabular models while enforcing the feature constraints. We show that CAPGD outperforms the other gradient-based attacks in terms of capability to generate valid (constraint-satisfying) adversarial examples.

2. We propose a new efficient and effective evasion attack (CAA) that combines gradient and search attacks to optimize both effectiveness and computational cost.

3. We evaluate CAA in a large-scale evaluation over four datasets, five architectures, and two training methods (standard and adversarial training). Our results show that CAA outperforms all other attacks and is up to 5 times more efficient.

## 2 Related work

### 2.1 Tabular Deep Learning

Tabular data remains the most commonly used form of data [34], especially in critical applications such as medical diagnosis [38, 36], financial applications [18, 11, 8], user recommendation systems [46], cybersecurity [9, 1], and more. Improving the performance and robustness of tabular

Table 1: Evasion attacks for tabular machine learning. Attacks with a public implementation in bold.

| Attack | Supported features | Supported constraints | | |
| --- | --- | --- | --- | --- |
| | | Categorical | Discrete | Relations |
| **LowProFool (LPF)** [3] | Continuous | No | No | No |
| Cartella et al. [8] | Continous, Discrete, Categorical | Yes | Yes | No |
| Gressel et al. [19] | Continous, Discrete, Categorical | Yes | Yes | No |
| Xu et al. [41] | Categorical | Yes | No | No |
| **Wang et al. [40]** | Categorical | Yes | No | No |
| **Bao et al. [4]** | Categorical | Yes | No | No |
| **BF*/BFS** [26, 25] | Continous, Discrete, Categorical | Yes | Yes | No |
| Mathov et al. [30] | Continous, Discrete, Categorical | Yes | Yes | No |
| **CPGD, MOEVA** [35] | Continous, Discrete, Categorical | Yes | Yes | Yes |
| **CAPGD, CAA (OURS)** | Continous, Discrete, Categorical | Yes | Yes | Yes |

machine learning models for these applications is becoming critical as more ML-based solutions are cleared to be deployed in critical settings.

Borisov et al. [5] showed that traditional deep neural networks tend to yield less favorable results in handling tabular data when compared to more shallow machine learning methods, such as XGBoost. However, recent approaches like RLN [33] and TabNet [2] are catching up and even outperforming shallow models in some settings. We argue that DNNs for Tabular Data are sufficiently mature and competitive with shallow models and require therefore a thorough investigation of their safety and robustness. Our work is the first exhaustive study of these critical properties.

## 2.2 Realistic Adversarial Examples

Initially applied to computer vision, adversarial examples have also been adapted and evaluated on tabular data. Ballet et al. [3] considered feature importance to craft the attacks, Mathov et al. [30] considered mutability, type, boundary, and data distribution constraints, Kireev et al. [24] suggested considering both the cost and benefit of perturbing each feature, and Simonetto et al. [35] introduced domain-constraints (relations between features) as a critical element of the attack.

While domain constraints satisfaction is essential for successful attacks, research on robustness for industrial settings (eg Ghamizi et al. [18] with a major bank) also demonstrated that imperceptibility remains important for critical systems with human-in-the-loop mechanisms, which could deflect attacks with manual checks from human operators. Imperceptibility is domain-specific, and multiple approaches have been suggested [3, 24, 16]. None of these approaches was confronted with human assessments or compared with each other, and in our study, we decided to use the most established $L_2$ norm. Our algorithms and approaches are generalizable to further distance metrics and imperceptibility definitions.

Overall, except the work from Simonetto et al. [35], none of the existing attacks for tabular machine learning supports the feature relationships inherent to realistic tabular datasets, as summarized in Table 1. Nevertheless, we evaluate all the approaches that support continuous values and where a public implementation is available to confirm our claims: LowProFool, BF*, CPGD, and MOEVA.

## 3 Problem formulation

We formulate the problem of evasion attacks under constraints. We assume the attack to be untargeted (i.e. it aims to force misclassification in any incorrect class); the formulation for targeted attacks is similar and omitted for space reasons.

We denote by $x \in \mathbb{R}^d$ an input example and by $y \in \{1, \ldots, C\}$ its correct label. Let $h : \mathbb{R}^d \to \mathbb{R}^C$ be a classifier and $h_{c_k}(x)$ the classification score that $h$ outputs for input $x$ to be in class $c_k$. Let $\Delta \subseteq \mathbb{R}^d$ be the space of allowed perturbations. Then, the objective of an evasion attack is to find a $\delta \in \Delta$ such that $argmax_{c \in \{1,\ldots,C\}} h_c(x + \delta) \neq y$.

In image classification, the set $\Delta$ is typically chosen as the perturbations within some $l_p$-ball around $x$, that is, $\Delta_p = \{\delta \in \mathbb{R}^d, ||\delta||_p \leq \epsilon\}$ for a maximum perturbation threshold $\epsilon$. This restriction aims at preserving the semantics of the original input by assuming that small enough perturbations will yield images that humans perceive the same as the original images and would therefore classify the perturbed input into the same class (while the classifier predicts another class). This also guarantees that the example remains meaningful, that is, $x + \delta$ is not an image with random noise.

Tabular data are by nature different from images. They typically represent objects of the considered application domain (e.g. botnet traffic [10], financial transaction [18]). We denote by $\varphi : Z \to \mathbb{R}^d$ the feature mapping function that maps objects of the problem space $Z$ to a $d$-dimensional feature space defined by the feature set $F = \{f_1, f_2, ...f_d\}$. Each object $z \in Z$ must inherently respect some natural condition to be valid (to be able to exist in reality). In the feature space, these conditions translate into a set of constraints on the feature values, which we denote by $\Omega$. By construction, any input example $x$ obtained from a real-world object $z$ satisfies $\Omega$, noted $x \models \Omega$.

Thus, in the case of tabular data, we additionally require the perturbation $\delta$ applied to $x$ to yield a valid example $x + \delta$ satisfying $\Omega$, that is, $\Delta_p(x) = \{\delta \in \mathbb{R}^d : ||\delta||_p \leq \epsilon \wedge x + \delta \models \Omega\}$.

To define the constraint language expressing $\Omega$, we consider the four types of constraint introduced by Simonetto et al. [35]. These four constraint types cover all the constraints of the datasets in our empirical study. Hence, *immutability* defines what features cannot be changed by an attacker; *boundaries* defines upper / lower bounds for feature values; *type* specifies a feature to take continuous, discrete, or categorical values; and *feature relationships* capture numerical relations between features. Feature relationship constraints can be expressed with the following grammar:

$$\omega := \omega_1 \wedge \omega_2 \mid \omega_1 \vee \omega_2 \mid \psi_1 \succeq \psi_2 \tag{1}$$

$$\psi := c \mid f_i \mid \psi_1 \oplus \psi_2 \mid x_i \tag{2}$$

Equation 1 means that a constraint formula $\omega$ can either be an intersection ($\wedge$), or a union ($\vee$) of two other constraint formulae $\omega_1, \omega_2$, or $\omega$ can be a comparison operator $\succeq \in \{<, \leq, =, \neq, \geq, >\}$ between two values $\psi_1$ and $\psi_2$.

Equation 2 details the numeric expressions that are supported by the grammar. A numeric expression $\psi$ can be constant $c$, an operation $\oplus \in \{+, -, *, /\}$ between two other numerical expressions $\psi_1$ and $\psi_2$, or a specific feature $f_i$, or the $i$-th feature of the clean sample. The difference between $f_i$ and $x_i$ is that $f_i$ corresponds to the current value of the evaluated example and $x_i$ corresponds to its original value in the clean example.

Let's consider one complex constraint from the LCLD credit scoring use case: the term of the loan can only be 36 or 60 months and the number of open accounts is lower than the number of allowed accounts for this client. Such a constraint can be formally written as:

$$\omega_1 = ((f_\text{term} = 36) \vee (f_\text{term} = 60)) \wedge (f_\text{open\_acc} \leq f_\text{total\_acc}) \tag{3}$$

.

We provide other examples in Appendix A.1.

### 3.1 Constrained Projected Gradient Descent

Constrained Projected Gradient Descent (CPGD) is an extension of the PGD attack [29] to generate adversarial examples satisfying constraints in tabular machine learning. It integrates constraint satisfaction into the loss function that PGD optimizes. This is achieved by translating each constraint $\omega$ into a differentiable function $penalty(x, \omega)$ that values to zero if $x \models \omega$; otherwise, the function represents how far $x$ is from satisfying $\omega$. We follow the definition of Table 5 in Appendix A.1 to translate each construct of the constraints grammar into a penalty function.

For instance, the penalty function of $\omega_1$ in Equation 3 is:

$$penalty(x, \omega_1) = min(|f_\text{term} - 36|, |f_\text{term} - 60|) + max(0, f_\text{open\_acc} - f_\text{total\_acc}) \tag{4}$$

Based on this, CPGD produces adversarial examples from an initial sample $x_{orig}$ classified as $y$ by iteratively computing:

$$x^{(k+1)} = R_\Omega\Big( P_\mathcal{S}\big(x^{(k)} + \eta^{(k)}\nabla\mathcal{L}(x^{(k)}, y, h, \Omega)\big)\Big) \tag{5}$$

where $x^0 = x_{orig}$ (the original input), $R_\Omega$ is a domain-specific repair operator [35], $P_\mathcal{S}$ is the projection onto $\mathcal{S} = \{x \in \mathbb{R}^d, ||x - x_{orig}||_p \le \epsilon\}$, $\nabla\mathcal{L}$ is the gradient of loss function $\mathcal{L}$, defined as

$$\mathcal{L}(x, y, h, \Omega) = l(h(x), y) - \sum_{\omega_i \in \Omega} penalty(x, \omega_i). \tag{6}$$

In the original CPGD implementation, the step size $\eta^{(k)}$ follows a predefined decay schedule, $\eta^{(k)} = \epsilon \times 10^{-(1+\lfloor k/\lfloor K/M\rfloor\rfloor)}$, with $M = 7$, and $K = max(k)$. $\mathcal{L}'(x)$ abbreviates $\mathcal{L}(x, y, h, \Omega)$.

## 3.2 Experimental settings

Our experiments are driven by the following datasets, models, and attack parameters. More details about the datasets and models are given in Appendix A.5.

**Datasets**   To conduct our study, we selected tabular datasets that present feature constraints from their respective application domain. **URL** [22] is a dataset of legitimate and phishing URLs. With only 14 linear domain constraints and 63 features, it is the simplest of our empirical study. **LCLD** [17] is a credit-scoring dataset with non-linear constraints. The **WiDS** [27] dataset contains medical data on the survival of patients admitted to the ICU. It has only 30 linear domain constraints. The **CTU** [10] dataset reports legitimate and botnet traffic from CTU University. The challenge of this dataset lies in its large number of linear domain constraints (360). We detail the datasets in the Appendix A.4.

**Architectures**   We evaluate five top-performing architectures from a recent survey on tabular ML [5]: **TabTransformer** [23] and **TabNet** [2] are transformer-based models. **RLN** [33] uses a regularization coefficient to minimize a counterfactual loss. **STG** [43] optimizes feature selection with stochastic gates, and **VIME** [44] relies on self-supervised learning. These architectures achieve performance equivalent to XGBoost, the best shallow machine learning model for our use cases.

**Perturbation parameters**   We use the L2-norm to measure the distance between original and perturbed inputs, because this norm is suitable for both numerical and categorical features. We set $\epsilon$ to 0.5 for all datasets. Each dataset has a critical (negative) class, respectively phishing URLs, rejected loans, flagged botnets, and not surviving patients. Hence, we only attack clean examples from the critical class that are not already misclassified by the model and report robust accuracy of models.

**Evaluation metrics**   We measure the effectiveness of our attack using robust accuracy defined as the accuracy of valid examples generated by a given attack. If a clean example is misclassified, we do not perturb it. If the attack generates an invalid example, we consider it as correctly classified. We measure the efficiency of the attacks in computational time.

# 4   Our Constrained *Adaptive* PGD

The relative lack of effectiveness of CPGD as reported in its original publication leads us to investigate the cause of these weaknesses. We investigate four factors that may affect the success rate of the attack: (1) we conjecture that the fixed step size and predefined decay in CPGD might be suboptimal because the choice of the step size is known to largely impact the effectiveness of gradient-based attacks [31]; (2) CPGD is unaware of the trend, i.e. it does not consider whether the optimization is evolving successfully and is not able to react to it; (3) CPGD does not check constraint satisfaction between the iterations, which could "lock" the algorithm into a part of the invalid data space; (4) CPGD starts with the original example, whereas classical gradient-based attacks often benefit from random initialization.

## 4.1 CAPGD algorithm

We propose Constrained Adaptive PGD (CAPGD), a new constraint-aware gradient-based attack that aims to overcome the limitations of CPGD and improve its effectiveness. We detail CAPGD in Algorithm 1 in Appendix A.2, and summarize its components below.

**Step size selection**    We introduce a step-size adaptation. We follow the exploration-exploitation principle by gradually reducing the gradient step [13]. However, unlike CPGD, this reduction does not follow a fixed schedule but is determined by the optimization trend. If the value of the loss function grows, we keep the same step size; otherwise, we halve it. That is, we start with a step $\eta^{(0)} = 2\epsilon$, and we identify checkpoints $w_0 = 0, w_1, ..., w_n$ at which we decide whether it is necessary to halve the size of the current step. We halve the step size if any of the following two conditions holds:

1. Since the last checkpoint, we count how many cases since the last checkpoint $w_{j-1}$ the update step has successfully increased $\mathcal{L}'$. The condition holds if the loss has increased for at least a fraction of $\rho$ steps (we set $\rho = 0.75$):

$$\sum_{i=w_{j-1}}^{w_j-1} \mathbf{1}_{\mathcal{L}'(x^{(i+1)})>\mathcal{L}'(x^{(i)})} < \rho \cdot (w_j - w_{j-1}). \tag{7}$$

2. The step has not been reduced at the last checkpoint and the loss is less or equal to the loss of the last checkpoint:

$$\eta^{(w_{j-1})} \equiv \eta^{(w_j)} \wedge \mathcal{L}_{\max}^{(w_{j-1})} \equiv \mathcal{L}_{\max}^{(w_j)} \tag{8}$$

where $\mathcal{L}'(x)$ is the loss function, $\mathcal{L}_{\max}^{(w_j)}$ is the highest loss value in the first $j + 1$ iterations.

**Repair operator**    While equality constraints are included in the penalty function, optimization alone does not achieve exact equality of feature values. Our new repair operator $R_\Omega$ improved from [35] addresses this by setting the value of the left-hand side of an equation of the form $f_i = \psi$ to match the evaluation of the right-hand side in each iteration. It maintains other dataset constraints such as bounds, mutability, and feature types but does not ensure other relational constraints are met. The operator can violate maximum perturbation constraints, yet at each iteration, the perturbation is corrected back within the allowed maximum. This approach has been shown to improve the success rate of CAPGD, as demonstrated by our ablation study in Table 8. We provide the algorithm in Appendix A.2.

**Initial state**    As for initialization, we apply the attack from two initial states: the original example $x_{orig}$ and a random example sampled from $\mathcal{S}$ (the Lp-ball around $x_{orig}$). The goal behind this second initialization is to reduce the risk of being immediately locked into local optima that encompass only invalid examples. Our experiments later reveal the complementary of these two initializations.

**Gradient step**    Finally, we introduce in CAPGD a momentum [14]. Let $\eta^{(k)}$ be the step size at iteration $k$, then we first compute $z^{(k+1)}$ before the updated example $x^{(k+1)}$.

$$z^{(k+1)} = P_{\mathcal{S}}\big(x^{(k)} + \eta^{(k)}(\nabla\mathcal{L}'(x^{(k)}))\big) \tag{9}$$
$$x^{(k+1)} = R_\Omega\Big(P_{\mathcal{S}}\big(x^{(k)} + \alpha \cdot (z^{(k+1)} - x^{(k)}) + (1 - \alpha) \cdot (x^{(k)} - x^{(k-1)})\big)\Big)$$

where $\alpha \in [0, 1]$ (we use $\alpha = 0.75$ following [13]) regulates the influence of the previous update on the current, and $P_{\mathcal{S}}$ is the projection onto $\mathcal{S} = \{x \in \mathbb{R}^d, ||x - x_{orig}||_p \leq \epsilon\}$.

## 4.2    Comparison of CAPGD to gradient-based attacks

To evaluate the benefits of CAPGD, we compare it with CPGD as well as LowProFool, to the best of our knowledge, the only other public *gradient* attack for tabular models that can be extended to support all types of features.

**CAPGD is more successful than existing gradient attacks.**    In Table 2, we compare the robust accuracy across our four datasets and five architectures with CPGD, LowProFool, and CAPGD. CAPGD significantly outperforms CPGD and LowProFool. It decreases the robust accuracy on URL, LCLD, and WIDS datasets to as low as 10.9%, 0.2%, and 10.2% respectively.

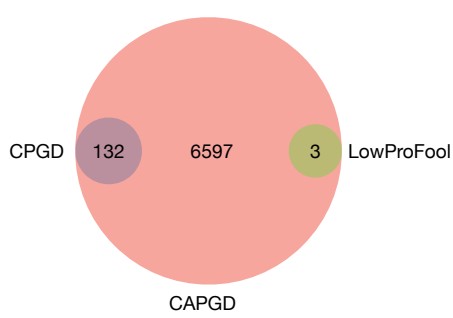

Figure 1: Visualization of the complementarity of CAPGD, CPGD, and LowProFool with the number of successful adversarial examples.

The results also reveal that gradient attacks are ineffective on the CTU dataset. These results demonstrate that gradient-based attacks are not enough and motivate us to consider combining CAPGD with search-based attacks, as investigated in Section 5.

Table 2: Robust accuracy against CAPGD and SOTA gradient attacks. A lower robust accuracy means a more effective attack (lowest in bold).

| DS | Model | Clean | LPF | CPGD | CAPGD |
|---|---|---|---|---|---|
| | TabTr. | 93.6 | 93.6 | 91.9 | **10.9** |
| | RLN | 94.4 | 94.4 | 92.8 | **12.6** |
| URL | VIME | 92.5 | 92.5 | 90.7 | **56.3** |
| | STG | 93.3 | 93.3 | 93.3 | **72.6** |
| | TabNet | 93.4 | 93.4 | 88.5 | **19.3** |
| | TabTr. | 69.5 | 69.2 | 69.5 | **27.1** |
| | RLN | 68.3 | 68.3 | 68.3 | **0.2** |
| LCLD | VIME | 67.0 | 67.0 | 67.0 | **2.6** |
| | STG | 66.4 | 66.4 | 66.4 | **55.5** |
| | TabNet | 67.4 | 67.4 | 67.4 | **6.3** |
| | TabTr. | **95.3** | **95.3** | **95.3** | **95.3** |
| | RLN | **97.8** | **97.8** | **97.8** | **97.8** |
| CTU | VIME | **95.1** | **95.1** | **95.1** | **95.1** |
| | STG | **95.3** | **95.3** | **95.3** | **95.3** |
| | TabNet | **96.1** | **96.1** | **96.1** | **96.1** |
| | TabTr. | 75.5 | 75.5 | 75.2 | **48.0** |
| | RLN | 77.5 | 77.5 | 77.3 | **61.8** |
| WIDS | VIME | 72.3 | 72.3 | 71.5 | **51.4** |
| | STG | 77.6 | 77.6 | 77.5 | **65.1** |
| | TabNet | 79.7 | 79.7 | 76.0 | **10.2** |

**CAPGD subsumes all gradient attacks.** We analyze in detail the original examples from which attacks could generate valid and successful adversarial examples. For each attack, we take the union of the sets of clean examples across 5 seeds. We generate the Venn diagram for CPGD, LowProFool, and CAPGD, for all datasets and model architectures. We sum the partition values in Figure 1. CAPGD generates adversarial examples for 6597 original examples from which none of the other gradient attacks could produce adversarial examples. In contrast, all successful adversarial examples by CPGD (132) and LowProFool (3) are also generated by CAPGD.

**All components of CAPGD contribute to its effectiveness.** We analyze in detail each of the components of CAPGD in Appendix B.1. These results and ablation studies confirm the complementarity of our new mechanisms and their contribution to the effectiveness of CAPGD.

## 5 CAA: an ensemble of gradient and search attacks

We next propose *Constrained Adaptive Attack* (CAA), an effective and efficient ensemble of gradient- and search-based attacks. The idea underlying CAA is that gradient-based attacks for tabular data are more efficient but less successful than search-based attacks. Thus, CAA integrates the best search-based attacks from each family in a complementary way, such that we maximize the set of adversarial examples that can be generated.

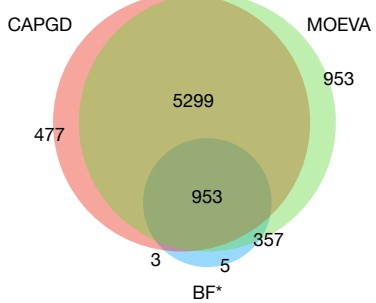

Figure 2: Visualization of the complementarity of CAPGD, MOEVA, and BF* with the number of successful adversarial examples.

### 5.1 Design of CAA

Following our related work study, we consider the search attacks MOEVA and BF* Kulynych et al. [26], Kireev et al. [25]. As a first step, we compare in Figure 2 these two attacks in terms of the original examples for which they could generate successful adversarial examples. We also include CAPGD in this comparison, since we have shown that this attack subsumes the other gradient-based attacks. Our results reveal that CAPGD and MOEVA together subsume BF* except for 5 examples. Additionally, CAPGD and MOEVA are complementary, with CAPGD generating 477 unique examples and MOEVA 953. Overall, the combination of CAPGD with MOEVA yields the strongest method including only one gradient-based

Table 3: Robust accuracy and attack duration for CAPGD, MOEVA, and CAA. The Clean column corresponds to the accuracy of the model on the subset of clean samples that we attack. A lower robust accuracy means a more effective attack. The lowest robust accuracy is in bold. A lower duration is better. The lowest time between MOEVA and CAA is in bold.

| Dataset | Model | Clean | Robust accuracy ($\downarrow$) | | | | Duration in seconds ($\downarrow$) | | | |
| | | | CAPGD | BF* | MOEVA | CAA | CAPGD | BF* | MOEVA | CAA |
|---|---|---|---|---|---|---|---|---|---|---|
| URL | TabTr. | 93.6 | $10.9_{\pm0.1}$ | $93.2_{\pm0}$ | $18.2_{\pm0.8}$ | $\mathbf{8.9_{\pm0.2}}$ | $1_{\pm0}$ | $33_{\pm0}$ | $75_{\pm2}$ | $\mathbf{17_{\pm1}}$ |
| | RLN | 94.4 | $12.6_{\pm0.2}$ | $93.8_{\pm0}$ | $23.6_{\pm0.5}$ | $\mathbf{10.8_{\pm0.2}}$ | $1_{\pm0}$ | $27_{\pm0}$ | $74_{\pm3}$ | $\mathbf{19_{\pm1}}$ |
| | VIME | 92.5 | $56.3_{\pm0.1}$ | $92.2_{\pm0}$ | $56.5_{\pm0.9}$ | $\mathbf{49.5_{\pm0.5}}$ | $2_{\pm1}$ | $32_{\pm0}$ | $70_{\pm1}$ | $\mathbf{51_{\pm2}}$ |
| | STG | 93.3 | $72.6_{\pm0.0}$ | $93.2_{\pm0}$ | $58.2_{\pm0.9}$ | $\mathbf{58.0_{\pm0.8}}$ | $2_{\pm0}$ | $58_{\pm0}$ | $90_{\pm0}$ | $\mathbf{73_{\pm3}}$ |
| | TabNet | 93.4 | $19.3_{\pm0.6}$ | $90.9_{\pm0}$ | $17.5_{\pm0.6}$ | $\mathbf{11.0_{\pm0.5}}$ | $8_{\pm0}$ | $444_{\pm0}$ | $165_{\pm4}$ | $\mathbf{58_{\pm1}}$ |
| LCLD | TabTr. | 69.5 | $27.1_{\pm0.9}$ | $61.1_{\pm0}$ | $10.7_{\pm0.8}$ | $\mathbf{7.9_{\pm0.6}}$ | $5_{\pm1}$ | $154_{\pm0}$ | $124_{\pm5}$ | $\mathbf{83_{\pm2}}$ |
| | RLN | 68.3 | $0.2_{\pm0.1}$ | $38.9_{\pm0}$ | $0.8_{\pm0.2}$ | $\mathbf{0.0_{\pm0.0}}$ | $1_{\pm0}$ | $147_{\pm0}$ | $50_{\pm1}$ | $\mathbf{10_{\pm3}}$ |
| | VIME | 67.0 | $2.6_{\pm0.2}$ | $52.6_{\pm0}$ | $24.1_{\pm1.5}$ | $\mathbf{2.4_{\pm0.1}}$ | $1_{\pm0}$ | $149_{\pm0}$ | $49_{\pm2}$ | $\mathbf{13_{\pm1}}$ |
| | STG | 66.4 | $55.5_{\pm0.2}$ | $\mathbf{53.0_{\pm0}}$ | $55.4_{\pm0.2}$ | $53.6_{\pm0.1}$ | $3_{\pm0}$ | $191_{\pm0}$ | $60_{\pm2}$ | $\mathbf{57_{\pm2}}$ |
| | TabNet | 67.4 | $6.3_{\pm0.4}$ | $49.0_{\pm0}$ | $0.8_{\pm0.1}$ | $\mathbf{0.4_{\pm0.1}}$ | $4_{\pm0}$ | $754_{\pm0}$ | $68_{\pm2}$ | $\mathbf{23_{\pm0}}$ |
| CTU | TabTr. | 95.3 | $95.3_{\pm0.0}$ | $95.3_{\pm0}$ | $95.3_{\pm0.0}$ | $95.3_{\pm0.0}$ | $4_{\pm0}$ | $371_{\pm0}$ | $\mathbf{98_{\pm4}}$ | $110_{\pm5}$ |
| | RLN | 97.8 | $97.8_{\pm0.0}$ | $97.5_{\pm0}$ | $\mathbf{94.0_{\pm0.2}}$ | $\mathbf{94.0_{\pm0.2}}$ | $9_{\pm8}$ | $12_{\pm0}$ | $\mathbf{98_{\pm3}}$ | $112_{\pm4}$ |
| | VIME | 95.1 | $95.1_{\pm0.0}$ | $95.1_{\pm0}$ | $\mathbf{40.8_{\pm4.7}}$ | $\mathbf{40.8_{\pm4.7}}$ | $4_{\pm0}$ | $924_{\pm0}$ | $\mathbf{107_{\pm3}}$ | $116_{\pm2}$ |
| | STG | 95.3 | $95.3_{\pm0.0}$ | $95.3_{\pm0}$ | $95.3_{\pm0.0}$ | $95.3_{\pm0.0}$ | $5_{\pm0}$ | $548_{\pm0}$ | $\mathbf{105_{\pm3}}$ | $119_{\pm4}$ |
| | TabNet | 96.1 | $96.1_{\pm0.0}$ | $13.0_{\pm0}$ | $\mathbf{0.0_{\pm0.0}}$ | $\mathbf{0.0_{\pm0.0}}$ | $7_{\pm0}$ | $816_{\pm0}$ | $\mathbf{157_{\pm7}}$ | $182_{\pm4}$ |
| WIDS | TabTr. | 75.5 | $48.0_{\pm0.3}$ | $67.7_{\pm0}$ | $59.2_{\pm0.6}$ | $\mathbf{45.9_{\pm0.3}}$ | $3_{\pm0}$ | $440_{\pm0}$ | $65_{\pm3}$ | $\mathbf{49_{\pm1}}$ |
| | RLN | 77.5 | $61.8_{\pm0.3}$ | $77.0_{\pm0}$ | $67.7_{\pm0.3}$ | $\mathbf{60.9_{\pm0.2}}$ | $3_{\pm0}$ | $2520_{\pm0}$ | $52_{\pm1}$ | $\mathbf{49_{\pm2}}$ |
| | VIME | 72.3 | $51.4_{\pm0.3}$ | $71.2_{\pm0}$ | $59.4_{\pm0.5}$ | $\mathbf{50.3_{\pm0.2}}$ | $2_{\pm0}$ | $1406_{\pm0}$ | $48_{\pm2}$ | $\mathbf{41_{\pm1}}$ |
| | STG | 77.6 | $65.1_{\pm0.4}$ | $77.5_{\pm0}$ | $68.8_{\pm0.3}$ | $\mathbf{63.8_{\pm0.2}}$ | $3_{\pm0}$ | $1888_{\pm0}$ | $64_{\pm1}$ | $\mathbf{59_{\pm1}}$ |
| | TabNet | 79.7 | $10.2_{\pm0.3}$ | $73.1_{\pm0}$ | $13.9_{\pm0.4}$ | $\mathbf{5.3_{\pm0.4}}$ | $5_{\pm0}$ | $10472_{\pm0}$ | $77_{\pm4}$ | $\mathbf{25_{\pm1}}$ |

attack and one search-based attack. One could also include BF* for a slight increase in effectiveness, but this would come at the computational cost of running this attack in addition to the other two; our experimental results (presented below) actually reveal that BF* brings a substantial computational cost compared to CAPGD and MOEVA. Hence, we stick to CAPGD and MOEVA only.

The principle of CAA is thus to successively apply CAPGD and MOEVA, in that order. By applying CAPGD first, CAA has the opportunity to generate valid adversarial examples at low computational cost (benefiting from the performance of gradient attacks compared to search attacks). If CAPGD fails on an original example, CAA executes the slower but more effective MOEVA method.

## 5.2 Effectiveness and efficiency of CAA

We evaluated the effectiveness (robust accuracy) and efficiency (computation time) of CAA compared to the other methods. The hyperparameters of all attacks are fixed for all experiments (see Section A.6 of the appendix) and follow the recommendation given in their original paper.

Table 3 shows that CAA achieves the best performance in all cases but one: for STG model and LCLD dataset, BF* achieves a robust accuracy 0.6 percentage points lower than CAA – these are the unique original examples from which BF* could generate successful adversarial examples. Overall, CAA leads to a decrease in accuracy of up to 96.1%, 84.3%, and 21.7% compared to CAPGD, BF*, and MOEVA respectively.

The main advantage of CAA is its ability to find "easy" constrained adversarial using the cheaper gradient attack CAPGD, before processing harder examples with expensive search. We compare in the right panel of Table 3 the cost of running each attack, i.e., its execution time. CAA shines particularly in terms of efficiency. CAA reduces execution costs by up to 5 times compared to MOEVA. It is significantly faster than MOEVA for LCLD, URL, and WIDS datasets (except STG), and marginally more costly for CTU.

CAA is up to 418 times faster than BF*. In particular, in the only case where BF* marginally outperforms CAA, BF* requires 3.4 times more computation to generate the adversarial examples.

In Appendix B.5, we evaluate our attack on shallow models in direct and transferability settings.

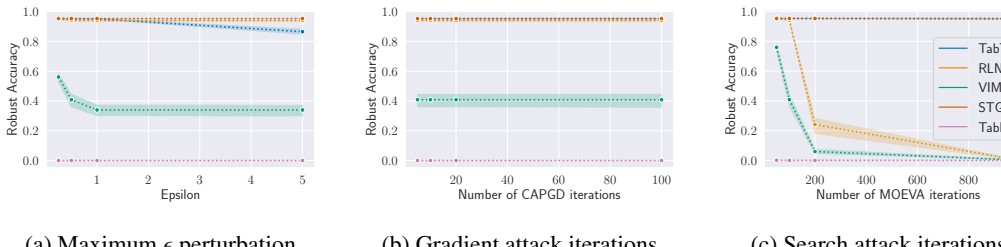

(a) Maximum $\epsilon$ perturbation     (b) Gradient attack iterations     (c) Search attack iterations

Figure 3: Impact of CAA budget on the robust accuracy for CTU dataset.

Table 4: CAA performances (XX+/-YY) against Madry adversarially trained model. XX refers to accuracy. YY is the difference between the accuracy of the adversarially trained model and standard training (cf. Table 3), such that '+' means a higher accuracy for the adversarially trained model.

| Dataset | Accuracy | TabTr. | RLN | VIME | STG | TabNet |
|---|---|---|---|---|---|---|
| URL | Clean | $93.9_{+0.3}$ | $95.2_{+0.8}$ | $93.4_{+0.9}$ | $94.3_{+1.0}$ | $99.5_{+6.1}$ |
| | CAA | $56.7_{+47.8}$ | $56.2_{+45.4}$ | $69.8_{+20.3}$ | $90.0_{+32.0}$ | $91.8_{+80.8}$ |
| LCLD | Clean | $73.9_{+4.4}$ | $69.5_{+1.2}$ | $65.5_{-1.5}$ | $15.6_{-50.8}$ | $0.0_{-67.4}$ |
| | CAA | $70.3_{+62.4}$ | $63.0_{+63.0}$ | $10.4_{+8.0}$ | $12.1_{-41.5}$ | $0.0_{-0.4}$ |
| CTU | Clean | $95.3_{+0.0}$ | $97.3_{-0.5}$ | $95.1_{+0.0}$ | $95.1_{-0.2}$ | $0.2_{-95.8}$ |
| | CAA | $95.3_{+0.0}$ | $97.1_{+3.0}$ | $94.0_{+53.2}$ | $95.1_{-0.2}$ | $0.2_{+0.2}$ |
| WIDS | Clean | $77.3_{+1.8}$ | $78.0_{+0.5}$ | $72.1_{-0.2}$ | $62.6_{-15.1}$ | $98.4_{+18.6}$ |
| | CAA | $65.1_{+19.2}$ | $66.6_{+5.7}$ | $52.1_{+1.8}$ | $45.2_{-18.6}$ | $58.4_{+53.1}$ |

CAPGD and MOEVA fail to generate adversarial examples on CTU for 2 out of 5 models. In Appendix B.3, we provide a possible explanation for this dataset by evaluating our attacks on different sub-sets of constraints with varying complexity.

## 5.3 Impact of attack budget

We study the impact of the attacker's budget on the effectiveness of CAA, in terms of (i) maximum perturbation $\epsilon$ and (ii) the number of iterations of its components. We focus on the CTU dataset, which models are the only ones to remain robust through our previous experiments. All the datasets are evaluated in Appendix B.2. Figure 3 reveals in (a) that the maximum perturbation distance $\epsilon$ has little impact on the effectiveness of the attack. Increasing the number of iterations for the gradient attack component (b) does not have an impact on the success rate of CAA. Increasing the budget of the search attack component (c) significantly impacts the robustness of some models. While TabTransformer and STG remain robust, the robust accuracy of RLM and VIME drops below 0.4 when doubling the number of search iterations to 200, and to zero with 1000 search iterations.

## 5.4 Impact of Adversarial training

We evaluate the effectiveness of our attack against models made robust with Madry's adversarial training (AT) [29], using examples generated by the PGD attack. We consider this defense because adversarial training-based methods were shown to be the only reliable defense against evasion attacks [37, 6]. In Table 4, we show the clean accuracy and the robust accuracy (against CAA) of the adversarially trained models (big numbers in Table 4). We also show the accuracy difference with the models trained with standard training (small numbers). In Appendix B.4, we evaluate additional defenses based on adversarial training and data augmentations.

**Adversarial training can degrade clean and robust performance.** As a preliminary check, we investigate whether adversarial training degrades the clean performance of the models. This is important to ensure that a non-increase of robust accuracy does not originate from clean performance degradation (instead of being due to CAA's strength). Our evaluation shows that adversarial training significantly degrades clean performance of the STG and Tabnet architectures. The accuracy of STG models drops to 15.6% and 62.6% for LCLD and WIDS respectively. As for Tabnet models, the clean accuracy drops to 0.0% (LCLD) and 0.2% (CTU). In all other cases, clean accuracy remains stable.

**CAA remains effective against adversarial training for some architectures.** The effectiveness of CAA against robust models is architecture- and dataset-dependent. The attack remains effective on VIME architecture applied to LCLD and WIDS, with robust accuracy as low as 10.1% and 52.2% respectively, as well as on RLN architecture on the WIDS dataset (66.6% robust accuracy and only +5.7% improvement compared to standard training). However, the robustness against CAA of Tabtransformer architecture is significantly improved on URL and LCLD datasets by respectively 47.8% and 62.4%, and marginally improved on WIDS dataset by 19.2%. Similarly, RLN robustness to CAA improves on URL (+45.4%) and LCLD (+63%).

# 6 Limitations

We identify three main limitations of our work.

*Marginal overhead of CAA:* In scenarios where CAPGD struggles to attack tabular models, CAA CAA can exhibit a computation overhead (<14%) compared to MOEVA. However, in 4 out of 5 evaluated settings, CAA is faster than MOEVA (up to 5 times).

*CAPGD effectiveness with complex constraints:* CAPGD effectiveness drops when increasing the constraint's complexity in the number of constraints or the number of features involved in each constraint.

*Coherence of constraints:* The mechanisms of CAA assume that the constraints definitions are sound. Incoherences between boundary constraints and feature relation constraints can lead to invalid adversarial examples with large $\epsilon$ budgets.

# 7 Broader Impact

Our work proposes CAA, the most effective evasion attack against constrained tabular ML. We also provided for each dataset at least one combination of architecture combined with AT where CAA can be mitigated. We expect that our work will have a more positive impact by leading to improved defenses in the scarcely explored field of robust tabular ML.

# 8 Conclusion

In this work, we first propose CAPGD, a new parameter-free gradient attack for constrained tabular machine learning. We also design CAA, a new Constrained Adaptive Attack that combines the best gradient-based attack (CAPGD) and the best search-based attack (MOEVA). We evaluate our attacks over four datasets and five architectures and demonstrated that our new attacks outperform all previous attacks in terms of effectiveness and efficiency. We believe that our work is a springboard for further research on the robustness of tabular machine learning and to open multiple research perspectives on constrained tabular ML. We hope that CAA will contribute to a faster development of adversarial defenses and recommend it as part of a standard evaluation pipeline of new tabular machine models.

## Acknowledgments and Disclosure of Funding

This research was funded in whole, or in part, by the Luxembourg National Research Fund (FNR), grant reference NCER22/IS/16570468/NCER-FT and grant BRIDGES/2022/IS/17437536. This research was supported by BGL BNP Paribas Luxembourg. In particular, we would like to thank Anne Goujon and Andrey Boytsov for their support with the financial use case. The experiments presented in this paper were carried out using the HPC facilities of the University of Luxembourg [39] (see `hpc.uni.lu`).

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

Table 5: From constraint formulae to penalty functions. $\tau$ is an infinitesimal value. *term*, *open_acc*, *total_acc*, *rec_per_month*, *record*, *month* are features and instances of $f$ in the grammar. 60 and 36 are constants and instances of $c$ in the grammar.

| Constraint | Penalty function | Constraint example | Penalty function example |
|---|---|---|---|
| $\psi_1 = \psi_2$ | $\mid \psi_1 - \psi_2 \mid$ | rec_per_month = record/month | $\mid$ rec_per_month $-$ (record/month) $\mid$ |
| $\psi_1 \leq \psi_2$ | $max(0, \psi_1 - \psi_2)$ | open_acc $\leq$ total_acc | $max(0, \text{open\_acc} - \text{total\_acc})$ |
| $\psi_1 < \psi_2$ | $max(0, \psi_1 - \psi_2$ $+\tau)$ | open_acc $<$ total_acc | $max(0, \text{open\_acc} - \text{total\_acc}$ $+10^{-5})$ |
| $\omega_1 \wedge \omega_2$ | $\omega_1 + \omega_2$ | $((\text{term} = 36) \vee (\text{term} = 60)) \wedge$ (open_acc $\leq$ total_acc) | $min(\mid \text{term} - 36 \mid, \mid \text{term} - 60 \mid) +$ $max(0, \text{open\_acc} - \text{total\_acc})$ |
| $\omega_1 \vee \omega_2$ | $min(\omega_1, \omega_2)$ | (term = 36) $\vee$ (term = 60) | $min(\mid \text{term} - 36 \mid, \mid \text{term} - 60 \mid)$ |

# Appendix

# A   Experimental protocol

## A.1   Constraints penalty function

The penalty function transforms each constraint formulation into a differentiable loss function to be minimized by gradient descent (or search algorithm). Table 5 inspired from [35] shows the supported constraints, their translation to penalty function, and examples for each supported constraint. Let's consider one complex constraint from the LCLD credit scoring use case: The term of the loan can only be 36 or 60 months and the number of open accounts is lower than the number of allowed accounts for this client. Such a constraint can be formally written as term $\in \{36, 60\}) \wedge$ (open_acc $\leq$ total_acc). The AND operator $\wedge$ is equivalent to a sum of losses, while the OR operator $\vee$ is described as $min(\mid c - a \mid, \mid c - b \mid, ...)$ in a loss function. Finally, the $a \leq b$ operator is equivalent to a $min(0, a - b)$ in a loss function. Hence, the complex constraint translates as the following penalty: $min(\mid term - 36 \mid, \mid term - 60 \mid) + max(0, \text{open\_acc} - \text{total\_acc})$.

## A.2   CAPGD Algorithm

Algorithm 1 summarizes the process of CAPGD. At each iteration, we compute the perturbation according to the definition of Equation (l. 7-9). We keep track of the best loss and accordingly best adversarial example found so far (l. 10-12). If we reach a checkpoint in $W$ and one of the conditions of Equation 7 or Equation 8 we half the step size $\eta$ (l. 13-17). We return the best adversarial example $x_{max}$.

Algorithm 2 describes the repair operator. For each constraint $\Omega$, if the constraint is an equality $=$ and the left operand $\omega.\text{left\_operand}$ is a feature (l. 3), we verify if the constraint is respected. If not $(penalty(x, \Omega) > 0)(l.4)$, we reset the value corresponding to feature $\omega.\text{left\_operand}$ in $x$ such that it equals the penalty function of the right operand $\omega.\text{right\_operand}$ (l. 5). According to the grammar Equation 1 and the penalty function definition in Table 5, the value feature $\omega.\text{left\_operand}$ must equal the penalty function of $\omega.\text{right\_operand}$ to satify the constraint.

## A.3   CAA Algorithm

Algorithm 3 summarizes the process of CAA. The algorithm takes as input the clean examples $X$, their associated labels $Y$, and the classifier $H$ such that $H(x) = argmax_{c \in \{1,...,C\}} h_c(x)$ (including its weights, loss function and probability function), $\Omega$ the set of domain constraints, and $\epsilon$ the maximum perturbation. We start by creating the mask of examples that are already adversarial (i.e. misclassified by the model) (l.3). We then split the clean examples that are already adversarial $X'$ (l.4), and the candidates $X_C$ on which we will execute the attacks. For each of our attacks (l.6), we generate a set of potentially adversarial examples from the candidate clean examples (l.7). Once again, we compute the mask of examples that are adversarial according to the subprocedure $is\_adv$ described below. According to the mask, we add the successful attack to the output $X'$ (l.8) and remove the associated clean examples from the candidate set $X_C$. Hence, for a given example, the

**Algorithm 1:** CAPGD

1: **Input:** $\mathcal{L}$, $h$ $S$, $\Omega$, $x^{(0)}$, y, $\eta$, $N_{\text{iter}}$, $W = \{w_0, \ldots, w_n\}$
2: **Output:** $x_{\max}$
3: $x^{(1)} \leftarrow P_{\mathcal{S}}\left(x^{(0)} + \eta \nabla \mathcal{L}'(x^{(0)})\right)$
4: $\mathcal{L}_{\max} \leftarrow \max\{\mathcal{L}'(x^{(0)}), \mathcal{L}'(x^{(1)})\}$
5: $x_{\max} \leftarrow x^{(0)}$ **if** $\mathcal{L}_{\max} \equiv \mathcal{L}'(x^{(0)})$ **else** $x_{\max} \leftarrow x^{(1)}$
6: **for** $k = 1$ **to** $N_{\text{iter}} - 1$ **do**
7: $\quad z^{(k+1)} \leftarrow P_{\mathcal{S}}\left(x^{(k)} + \eta \nabla \mathcal{L}'(x^{(k)})\right)$
8: $\quad x^{(k+1)} \leftarrow P_{\mathcal{S}}\left(x^{(k)} + \alpha(z^{(k+1)} - x^{(k)})\right.$
$$\left. + (1 - \alpha)(x^{(k)} - x^{(k-1)})\right)$$
9: $\quad x^{(k+1)} \leftarrow R_{\Omega}(x^{(k+1)})$
10: $\quad$ **if** $\mathcal{L}'(x^{(k+1)}) > \mathcal{L}_{\max}$ **then**
11: $\quad\quad x_{\max} \leftarrow x^{(k+1)}$ and $\mathcal{L}_{\max} \leftarrow \mathcal{L}'(x^{(k+1)})$
12: $\quad$ **end if**
13: $\quad$ **if** $k \in W$ **then**
14: $\quad\quad$ **if** Condition 1 **or** Condition 2 **then**
15: $\quad\quad\quad \eta \leftarrow \eta/2$
16: $\quad\quad$ **end if**
17: $\quad$ **end if**
18: **end for**

---

**Algorithm 2:** Constraint Repair for Input $x$ and Constraints Set $\Omega$

1: **Input:** $x, \Omega$
2: **for** each $\omega \in \Omega$ **do**
3: $\quad$ **if** $\omega$ is an EqualityConstraint **and** $\omega$.left_operand $\in F$ **then**
4: $\quad\quad$ **if** penalty$(x, \omega) > 0$ **then**
5: $\quad\quad\quad x_{\omega.\text{left\_operand}} \leftarrow$ penalty$(x, \omega.\text{right\_operand})$
6: $\quad\quad$ **end if**
7: $\quad$ **end if**
8: **end for**

---

**Algorithm 3:** CAA

1: **Input:** $X, Y, H, \Omega, \epsilon$
2: **Output:** $X'$
3: $adv\_mask = is\_adv(X, X, Y, H, \Omega, \epsilon)$
4: $X' \leftarrow X[adv\_mask]$
5: $X_c \leftarrow X[\neg adv\_mask], Y_c = Y[\neg adv\_mask]$
6: **for** $Attack$ **in** $\{CAPGD, MOEVA\}$ **do**
7: $\quad X_i \leftarrow Attack(X_c, Y_c, H, \Omega, \epsilon)$
8: $\quad adv\_mask \leftarrow is\_adv(X_i, X_c, Y_c, H, \Omega, \epsilon)$
9: $\quad X' \leftarrow X' \cup X_i[adv\_mask]$
10: $\quad X_c \leftarrow X_c[\neg adv\_mask], Y_c \leftarrow Y_c[\neg adv\_mask]$
11: **end for**
12: $X' \leftarrow X' \cup X_c$

13: SubProcedure **is_adv**$(X_i, X_c, Y_c, H, \Omega, \epsilon)$ :
14: $\quad adv\_mask \leftarrow \{\}$
15: $\quad$ **for** $k = 1$ **to** $N_{\text{iter}}|X|$ **do**
16: $\quad\quad adv \leftarrow (X_i[k] \models \Omega) \wedge (H(X_i[k]) \neq Y_c[k]) \wedge (L_p(X_i[k], X_c[k]) \leq \epsilon)$
17: $\quad\quad adv\_mask \leftarrow adv\_mask \cup adv$
18: $\quad$ **end for**
19: $\quad$ return $adv\_mask$

Table 6: The datasets evaluated in the empirical study, with the class imbalance of each dataset.

| Dataset | Task | Properties Size | # Features | Balance (%) |
|---------|------|------|-----------|-------------|
| LCLD [17] | Credit Scoring | 1 220 092 | 28 | 80/20 |
| CTU-13 [10] | Botnet Detection | 198 128 | 756 | 99.3/0.7 |
| URL [22] | Phishing URL detection | 11 430 | 63 | 50/50 |
| WIDS [27] | ICU patient survival | 91 713 | 186 | 91.4/8.6 |

next attack is only executed if no attack has been successful, reducing the overall cost of CAA. At the end of CAA, we had the remaining candidates, for which we have not found adversarial examples to the output set of potentially adversarial examples $X'$, to ease the calculation of robust accuracy (e.g. in transferable settings).

The sub procedure $is\_adv$ goes through all the examples $X_i[k] \in X_i$ and adds $True$ to the mask, if all of the following conditions hold:

- $X_i[k]$ respects the domain constraints,
- $X_i[k]$ classification by $H$ is different from its true label $Y_c[k]$,
- $X_i[k]$ pertubation w.r.t to $X_c[k]$ is lower or equal to $\epsilon$.

## A.4 Datasets

Our dataset design followed the same protocol as Simonetto et al.[35]. We present in Table 6 the attributes of our datasets and the test performance achieved by each of the architectures.

**Credit scoring - LCLD**   (licence: CC0: Public Domain) We engineer a dataset from the publicly available Lending Club Loan Data[1]. This dataset contains 151 features, and each example represents a loan that was accepted by the Lending Club. However, among these accepted loans, some are not repaid and charged off instead. Our goal is to predict, at the request time, whether the borrower will be repaid or charged off. This dataset has been studied by multiple practitioners on Kaggle. However, the original version of the dataset contains only raw data and to the extent of our knowledge, there is no featured engineered version commonly used. In particular, one shall be careful when reusing feature-engineered versions, as most of the versions proposed present data leakage in the training set that makes the prediction trivial. Therefore, we propose our own feature engineering. The original dataset contains 151 features. We remove the example for which the feature "loan status" is different from "Fully paid" or "Charged Off" as these represent the only final status of a loan: for other values, the outcome is still uncertain. For our binary classifier, a 'Fully paid" loan is represented as 0 and a "Charged Off" as 1. We start by removing all features that are not set for more than 30% of the examples in the training set. We also remove all features that are not available at loan request time, as this would introduce bias. We impute the features that are redundant (e.g. grade and sub-grade) or too granular (e.g. address) to be useful for classification. Finally, we use one-hot encoding for categorical features. We obtain 47 input features and one target feature. We split the dataset using random sampling stratified on the target class and obtained a training set of 915K examples and a testing set of 305K. They are both unbalanced, with only 20% of charged-off loans (class 1). We trained a neural network to classify accepted and rejected loans. It has 3 fully connected hidden layers with 64, 32, and 16 neurons.

For each feature of this dataset, we define boundary constraints as the extremum value observed in the training set. We consider the 19 features that are under the control of the Lending Club as immutable. We identify 10 relationship constraints (3 linear, and 7 non-linear ones).

**URL Phishing - ISCX-URL2016**   (license CC BY 4.0) Phishing attacks are usually used to conduct cyber fraud or identity theft. This kind of attack takes the form of a URL that reassembles a legitimate URL (e.g. user's favorite e-commerce platform) but redirects to a fraudulent website that asks the user for their personal or banking data. [22] extracted features from legitimate and fraudulent URLs

---

[1]https://www.kaggle.com/wordsforthewise/lending-club

Table 7: The three model architectures of our study.

| Family | Model | Hyperparameters |
|---|---|---|
| Transformer | TabTransformer | $hidden\_dim, n\_layers,$ $learning\_rate, norm, \theta$ |
| Transformer | TabNet | $n\_d, n\_steps,$ $\gamma, cat\_emb\_dim, n\_independent,$ $n\_shared, momentum, mask\_type$ |
| Regularization | RLN | $hidden\_dim, depth,$ $heads, weight\_decay,$ $learning\_rate, dropout$ |
| Regularization | STG | $hidden\_dims, learning\_rate, lam$ |
| Encoding | VIME | $p_m, \alpha, K, \beta$ |

as well as external service-based features to build a classifier that can differentiate fraudulent URLs from legitimate ones. The feature extracted from the URL includes the number of special substrings such as "www", "&", ",", "$", "and", the length of the URL, the port, the appearance of a brand in the domain, in a subdomain or in the path, and the inclusion of "http" or "https". External service-based features include the Google index, the page rank, and the presence of the domain in the DNS records. The complete list of features is present in the reproduction package. [22] provide a dataset of 5715 legit and 5715 malicious URLs. We use 75% of the dataset for training and validation and the remaining 25% for testing and adversarial generation.

We extract a set of 14 relation constraints between the URL features. Among them, 7 are linear constraints (e.g. length of the hostname is less or equal to the length of the URL) and 7 are Boolean constraints of the type $if\ a > 0$ then $b > 0$ (e.g. if the number of http $> 0$ then the number slash "/" $> 0$).

**Botnet attacks - CTU-13** (license CC BY NC SA 4.0) This is a feature-engineered version of CTU-13 proposed by [9]. It includes a mix of legit and botnet traffic flows from the CTU University campus. Chernikova et al. aggregated the raw network data related to packets, duration, and bytes for each port from a list of commonly used ports. The dataset is made of 143K training examples and 55K testing examples, with 0.74% examples labeled in the botnet class (traffic that a botnet generates). Data have 756 features, including 432 mutable features. We identified two types of constraints that determine what feasible traffic data is. The first type concerns the number of connections and requires that an attacker cannot decrease it. The second type is inherent constraints in network communications (e.g. maximum packet size for TCP/UDP ports is 1500 bytes). In total, we identified 360 constraints.

**WiDS** (license: PhysioNet Restricted Health Data License 1.5.0 [2]) [27] dataset contains medical data on the survival of patients admitted to the ICU. The goal is to predict whether the patient will survive or die based on biological features (e.g. for triage). This very unbalanced dataset has 30 linear relation constraints.

## A.5 Model architectures

Table 7 summarizes the family, model architecture, and hyperparameters tuned during the training of our models.

**TabTransformer** is a transformer-based model [23]. It uses self-attention to map the categorical features to an interpretable contextual embedding, and the paper claims this embedding improves the robustness of models to noisy inputs.

**TabNet** is another transformer-based model [2]. It uses multiple sub-networks that are used in sequence. At each decision step, it uses sequential attention to choose which features to reason. TabNet aggregates the outputs of each step to obtain the decision.

---

[2]https://physionet.org/content/widsdatathon2020/view-license/1.0.0/

**RLN** or Regularization Learning Networks [33] uses an efficient hyperparameter tuning scheme in order to minimize a counterfactual loss. The authors train a regularization coefficient to weights in the neural network in order to lower the sensitivity and produce very sparse networks.

**STG** or Stochastic Gates [43] uses stochastic gates for feature selection in neural network estimation problems. The method is based on probabilistic relaxation of the $l_0$ norm of features or the count of the number of selected features.

**VIME** or Value Imputation for Mask Estimation [44] uses self and then semi-supervised learning through deep encoders and predictors.

### A.6  Attacks parameters

For existing attacks, we reuse the hyperparameters proposed in their respective papers. For LowPro-Fool, we use a small step size of $\eta = 0.001$, $\lambda = 8.5$ rade off factor between fooling the classifier and generating imperceptible adversarial example, and run the attack for $N_{iter} = 20,000$ iterations. All other gradient attacks run for $N_{iter} = 10$ iterations. The schedule of decreasing steps of CPGD uses $M = 7$. In CAPGD, we fix the checkpoints as $w_j = \lceil p_j N_{\text{iter}} \rceil \leq N_{\text{iter}}$, with $p_j \in [0,1]$ defined as $p_0 = 0$, $p_1 = 0.22$ and

$$p_{j+1} = p_j + \max\{p_j - p_{j-1} - 0.03, 0.06\}.$$

. The influence of the previous update on the current update is set to $\alpha = 0.75$, and $\rho = 0.75$ for the halving of the step. MOEVA runs for $n_{gen} = 100$ iterations generating $n_{off} = 100$ offspring per iteration. Among the offspring, $n_{pop} = 200$ survive and are used for mating in the next iteration. With BF*, we discretize numerical features in $n_{bin} = 150$ bins. We run the attack for a maximum of $N_{iter} = 100$ iterations. CAA applies CAPGD and MOEVA with the same parameters.

### A.7  Hardware

We run our experiments on an HPC cluster node with 32 cores and 64GB of RAM dedicated to our task. Each node consists of 2 AMD Epyc ROME 7H12 @ 2.6 GHz for a total of 128 cores with 256 GB of RAM.

### A.8  Reproduction package and availability

The source code, datasets and pre-trained models to reproduce the experiments of this paper are available with the submission. The source code will be available publicly upon acceptance under the MIT license or similar.

## B  Additional results

### B.1  Components of CAPGD

**All components of CAPGD contribute to its effectiveness.** We conduct an ablation study on the components of CAPGD. We evaluate CAPGD without the repair operator at each iteration (NREP), without the initialization with clean example (NINI), without the initialization with random sampling (NRAN), and without the adaptive step (NADA). Table 8 reveals that removing a component of CAPGD reduces its effectiveness. Interestingly, not all components affect all datasets similarly. For instance, removing the repair at each gradient iteration only affects the LCLD datasets' success rate. For URL and WIDS, CAPGD-NREP remains in the confidence interval of CAPGD. Removing any other components always negatively affects CAPGD, up to an improvement of 32.1% of the robust accuracy for CAPGD-NADA on the WIDS dataset and TabNet model.

**CAPGD components are complementary.** None of the components of CAPGD negatively affects its capability of finding an adversarial example for a given clean example. In Figure 4, we analyze the coverage of each CAPGD variant A (Covering Attack) with regard to another variant B (Covered Attack). For A and B, we compute the set of clean examples $C_A$ and $C_B$ on which the attacks are

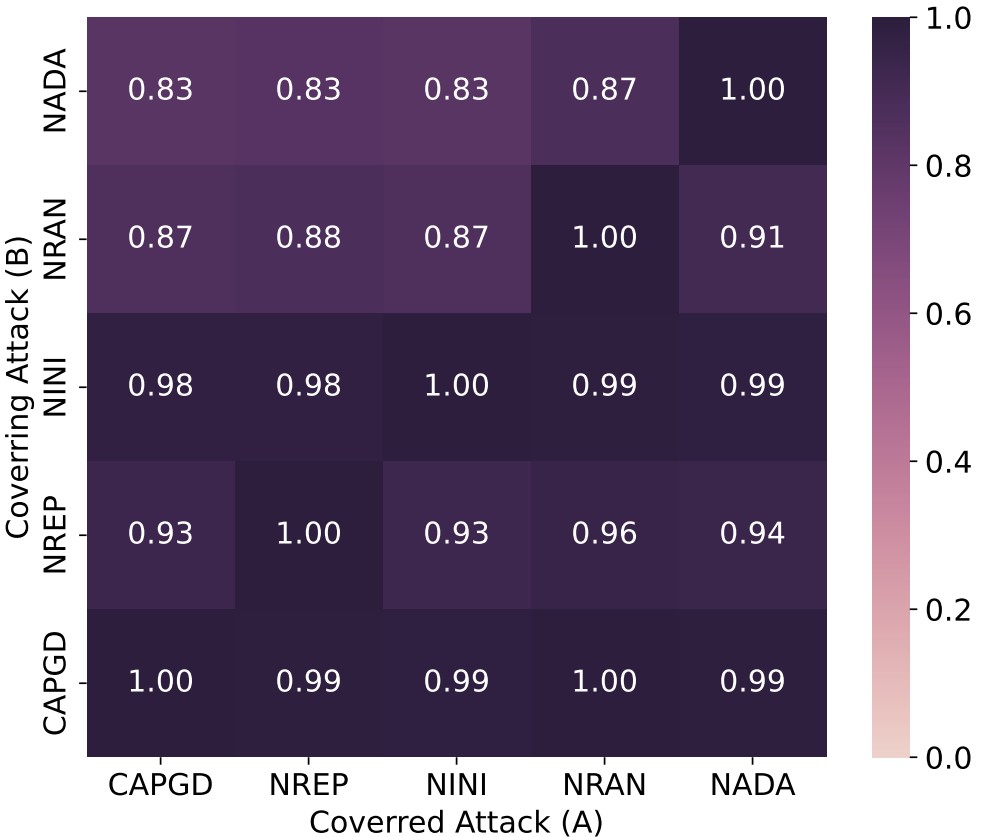

Figure 4: Visualization of the utility of CAA's components. For attack A (respectively B) we compute the set of clean examples $C_A$ (respectively $C_B$)) on which the attack is successful. The percentage represents the proportion of the set $C_A \cup C_B$ is covered by $C_B$. CAPGD-NADA is CAPGD without adaptive step, CAPGD-NRAN is CAPGD without the random start, CAPGD-NINI is CAPGD without the clean example initialization and CAPGD NREP is CAPGD without repair at each iteration.

successful. The percentage in the heatmaps represents the proportion of $C_A \cup C_B$ covered by $C_B$, that is

$$coverage = \frac{|C_B|}{|C_A \cup C_B|}$$

where $|C|$ is the cardinality of $C$. Attack B subsumes A when $coverage = 1$. In practice, to avoid random effect, we run the attack for $N = 5$ seeds, and take the *union* of clean examples on which we generate a successful example.

Figure 4 reveals that CAPGD subsumes all its variants with coverage over 99%, while none of the variants subsumes CAPGD. Therefore all components of CAPGD are necessary to obtain the strongest attack.

## B.2 Budget of attacker

In this section, we study the impact of CAA's budget on its effectiveness. We consider 3 budgets: the maximum perturbation $\epsilon$ allowed, the number of iterations in the gradient attack CAPGD (without changing MOEVA's budget), and the number of iterations in the search attack MOEVA (without changing CAPGD's budget).

Table 8: Ablation study: Robust accuracy for CAPGD and its variant without key components. The Clean column corresponds to the accuracy of the model on the subset of clean samples that we attack. A lower robust accuracy means a more effective attack. The lowest robust accuracy is in bold.

| Dataset | Model | Clean | NREP | NINI | NRAN | NADA | CAPGD |
|---|---|---|---|---|---|---|---|
| URL | TabTr. | 93.6 | $\mathbf{10.9}_{\pm 0.1}$ | $11.8_{\pm 0.3}$ | $12.6_{\pm 0.0}$ | $34.6_{\pm 0.4}$ | $\mathbf{10.9}_{\pm 0.1}$ |
| | RLN | 94.4 | $\mathbf{12.7}_{\pm 0.2}$ | $14.9_{\pm 0.2}$ | $14.8_{\pm 0.0}$ | $30.2_{\pm 0.5}$ | $\mathbf{12.6}_{\pm 0.2}$ |
| | VIME | 92.5 | $\mathbf{56.3}_{\pm 0.1}$ | $58.1_{\pm 0.3}$ | $56.9_{\pm 0.0}$ | $65.2_{\pm 0.1}$ | $\mathbf{56.3}_{\pm 0.1}$ |
| | STG | 93.3 | $\mathbf{72.6}_{\pm 0.0}$ | $73.4_{\pm 0.2}$ | $73.0_{\pm 0.0}$ | $75.3_{\pm 0.1}$ | $\mathbf{72.6}_{\pm 0.0}$ |
| | TabNet | 93.4 | $\mathbf{19.2}_{\pm 0.7}$ | $27.6_{\pm 0.8}$ | $29.7_{\pm 0.0}$ | $34.4_{\pm 0.3}$ | $19.3_{\pm 0.6}$ |
| LCLD | TabTr. | 69.5 | $38.3_{\pm 0.4}$ | $38.0_{\pm 0.9}$ | $38.0_{\pm 0.0}$ | $44.4_{\pm 1.1}$ | $\mathbf{27.1}_{\pm 0.9}$ |
| | RLN | 68.3 | $5.3_{\pm 0.2}$ | $1.4_{\pm 0.3}$ | $1.6_{\pm 0.0}$ | $1.1_{\pm 0.3}$ | $\mathbf{0.2}_{\pm 0.1}$ |
| | VIME | 67.0 | $17.9_{\pm 0.6}$ | $7.1_{\pm 0.5}$ | $7.3_{\pm 0.0}$ | $3.6_{\pm 0.4}$ | $\mathbf{2.6}_{\pm 0.2}$ |
| | STG | 66.4 | $59.4_{\pm 0.1}$ | $58.0_{\pm 0.3}$ | $56.5_{\pm 0.0}$ | $59.7_{\pm 0.2}$ | $\mathbf{55.5}_{\pm 0.2}$ |
| | TabNet | 67.4 | $30.4_{\pm 0.6}$ | $33.1_{\pm 1.4}$ | $10.8_{\pm 0.0}$ | $7.3_{\pm 0.4}$ | $\mathbf{6.3}_{\pm 0.4}$ |
| CTU | TabTr. | $\mathbf{95.3}$ | $\mathbf{95.3}_{\pm 0.0}$ | $\mathbf{95.3}_{\pm 0.0}$ | $\mathbf{95.3}_{\pm 0.0}$ | $\mathbf{95.3}_{\pm 0.0}$ | $\mathbf{95.3}_{\pm 0.0}$ |
| | RLN | $\mathbf{97.8}$ | $\mathbf{97.8}_{\pm 0.0}$ | $\mathbf{97.8}_{\pm 0.0}$ | $\mathbf{97.8}_{\pm 0.0}$ | $\mathbf{97.8}_{\pm 0.0}$ | $\mathbf{97.8}_{\pm 0.0}$ |
| | VIME | $\mathbf{95.1}$ | $\mathbf{95.1}_{\pm 0.0}$ | $\mathbf{95.1}_{\pm 0.0}$ | $\mathbf{95.1}_{\pm 0.0}$ | $\mathbf{95.1}_{\pm 0.0}$ | $\mathbf{95.1}_{\pm 0.0}$ |
| | STG | $\mathbf{95.3}$ | $\mathbf{95.3}_{\pm 0.0}$ | $\mathbf{95.3}_{\pm 0.0}$ | $\mathbf{95.3}_{\pm 0.0}$ | $\mathbf{95.3}_{\pm 0.0}$ | $\mathbf{95.3}_{\pm 0.0}$ |
| | TabNet | $\mathbf{96.1}$ | $\mathbf{96.1}_{\pm 0.0}$ | $\mathbf{96.1}_{\pm 0.0}$ | $\mathbf{96.1}_{\pm 0.0}$ | $\mathbf{96.1}_{\pm 0.0}$ | $\mathbf{96.1}_{\pm 0.0}$ |
| WIDS | TabTr. | 75.5 | $\mathbf{48.2}_{\pm 0.3}$ | $54.7_{\pm 0.9}$ | $53.3_{\pm 0.0}$ | $64.9_{\pm 0.6}$ | $\mathbf{48.0}_{\pm 0.3}$ |
| | RLN | 77.5 | $\mathbf{61.8}_{\pm 0.3}$ | $66.3_{\pm 0.4}$ | $63.7_{\pm 0.0}$ | $72.3_{\pm 0.5}$ | $\mathbf{61.8}_{\pm 0.3}$ |
| | VIME | 72.3 | $\mathbf{51.4}_{\pm 0.3}$ | $55.6_{\pm 0.9}$ | $54.1_{\pm 0.0}$ | $62.9_{\pm 0.4}$ | $\mathbf{51.4}_{\pm 0.3}$ |
| | STG | 77.6 | $\mathbf{65.1}_{\pm 0.4}$ | $68.9_{\pm 0.3}$ | $67.6_{\pm 0.0}$ | $74.0_{\pm 0.2}$ | $\mathbf{65.1}_{\pm 0.4}$ |
| | TabNet | 79.7 | $\mathbf{10.1}_{\pm 0.5}$ | $20.7_{\pm 0.9}$ | $17.5_{\pm 0.0}$ | $42.3_{\pm 0.7}$ | $10.2_{\pm 0.3}$ |

For each budget, we provide figures and detailed numerical results in tables, corresponding to the same experiment.

**Maximum perturbation** $\epsilon$    Figure 5 (numerical results in Table 9) reveals that increasing the maximum perturbation $\epsilon$ for CAA reduces the robust accuracy of the model in 16/20 cases.

**Number of CAPGD iterations**    Figure 6 (numerical results in Table 10) reveals that increasing the number of iterations for the gradient attack component has a limited impact on the success rate of CAA. The maximum drop of accuracy is 3.5% points between 10 and 100 iterations for WIDS/TabNet.

**Number of MOEVA iterations**    Figure 8 (numerical results in Table 11) reveals that increasing the number of iterations for the search attack component only reduces the robust accuracy in 4/20 cases (URL/VIME, URL/STG, CTU/VIME, and CTU/RLN).

We also observe that for TabTransformer and LCLD the robust accuracy increases with the number of search iterations. MOEVA is a multi-objective genetic algorithm. An inherent problem of multi-objective optimization is the trade-off between the objectives. If all solutions in the population are on the Pareto front, the algorithm must decide which solutions to discard for the next iteration, potentially discarding a valid adversarial example in our case.

Figure 7 shows the evolution of the success rate of MOEVA with the number of iterations in the same settings as in Figure 8b for TabTransformer. We find that the success rate reaches a maximum with 100 iterations. We argue that valid adversarial examples were discarded when the search continued to 1000 iterations. To confirm our hypothesis, we run the same experiment with a 10 times larger search population, such that more solutions are preserved at each iteration. We observe that in this setting, MOEVA converges slower (due to less selection pressure) but the success rate strictly increases with the number of generations. Increasing the population size also increases the execution time (by 3.4x in this case), due to the selection operator overhead.

Our approach CAA aims at minimizing the memory and computation overheads while maximizing the success rate, and CAA can be tuned to lead to lower robust accuracy with more iterations if the search space is expanded (for example with larger populations).

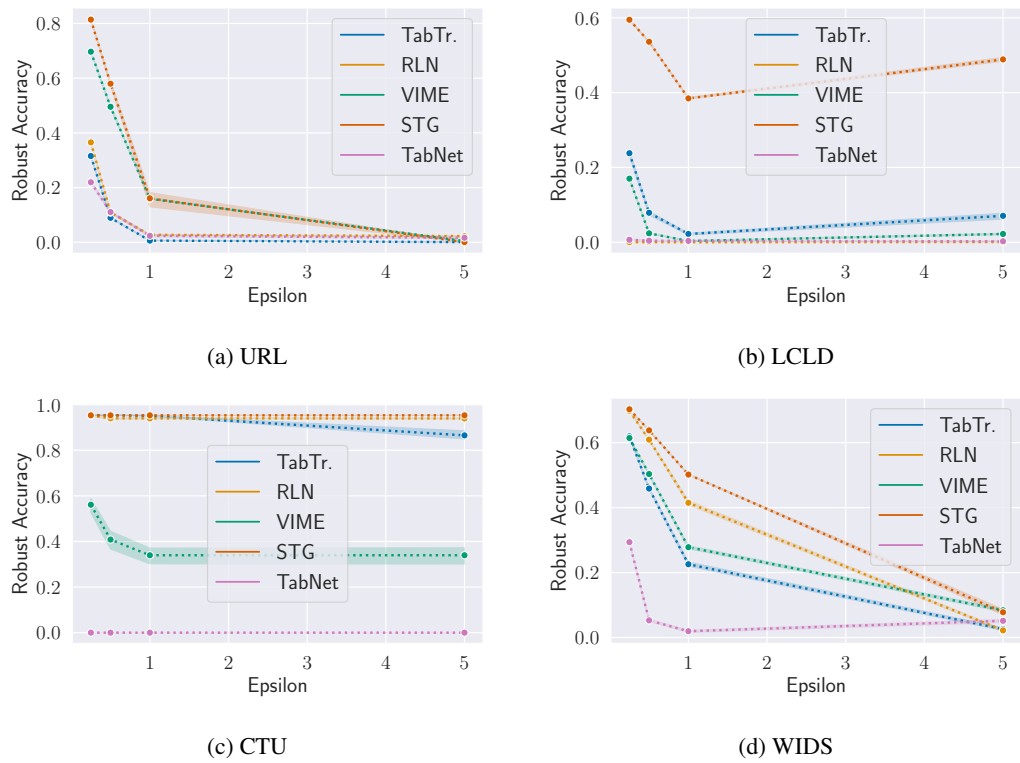

Figure 5: Robust accuracy with CAA with varying maximum perturbation $\epsilon$ budget.

Table 9: Robust accuracy with CAA with varying maximum perturbation $\epsilon$ budget. The lowest robust accuracy is in bold.

| Dataset | Model | Maximum perturbation $\epsilon$ | | | |
| | | 0.25 | 0.5 | 1.0 | 5.0 |
|---|---|---|---|---|---|
| URL | TabTr. | $31.5_{\pm 0.1}$ | $8.9_{\pm 0.2}$ | $0.6_{\pm 0.1}$ | $\mathbf{0.0_{\pm 0.0}}$ |
| | RLN | $36.5_{\pm 0.4}$ | $10.8_{\pm 0.2}$ | $2.7_{\pm 0.2}$ | $\mathbf{2.3_{\pm 0.0}}$ |
| | VIME | $69.7_{\pm 0.2}$ | $49.5_{\pm 0.5}$ | $15.9_{\pm 0.1}$ | $\mathbf{0.4_{\pm 0.3}}$ |
| | STG | $81.4_{\pm 0.2}$ | $58.0_{\pm 0.8}$ | $16.1_{\pm 3.1}$ | $\mathbf{0.0_{\pm 0.0}}$ |
| | TabNet | $21.9_{\pm 0.7}$ | $11.0_{\pm 0.5}$ | $\mathbf{2.3_{\pm 0.4}}$ | $1.6_{\pm 0.3}$ |
| LCLD | TabTr. | $23.8_{\pm 0.7}$ | $7.9_{\pm 0.6}$ | $\mathbf{2.2_{\pm 0.3}}$ | $7.1_{\pm 0.9}$ |
| | RLN | $\mathbf{0.0_{\pm 0.0}}$ | $\mathbf{0.0_{\pm 0.0}}$ | $\mathbf{0.0_{\pm 0.0}}$ | $0.1_{\pm 0.0}$ |
| | VIME | $17.0_{\pm 0.2}$ | $2.4_{\pm 0.1}$ | $\mathbf{0.3_{\pm 0.1}}$ | $2.2_{\pm 0.2}$ |
| | STG | $59.5_{\pm 0.3}$ | $53.6_{\pm 0.1}$ | $\mathbf{38.5_{\pm 0.2}}$ | $48.9_{\pm 0.6}$ |
| | TabNet | $0.7_{\pm 0.2}$ | $\mathbf{0.4_{\pm 0.1}}$ | $\mathbf{0.4_{\pm 0.1}}$ | $0.3_{\pm 0.1}$ |
| CTU | TabTr. | $95.3_{\pm 0.0}$ | $95.3_{\pm 0.0}$ | $95.1_{\pm 0.2}$ | $\mathbf{86.5_{\pm 1.9}}$ |
| | RLN | $95.2_{\pm 0.3}$ | $\mathbf{94.0_{\pm 0.2}}$ | $\mathbf{94.0_{\pm 0.2}}$ | $\mathbf{94.0_{\pm 0.2}}$ |
| | VIME | $56.1_{\pm 3.4}$ | $\mathbf{40.8_{\pm 4.7}}$ | $\mathbf{34.0_{\pm 4.0}}$ | $\mathbf{34.0_{\pm 4.0}}$ |
| | STG | $\mathbf{95.3_{\pm 0.0}}$ | $\mathbf{95.3_{\pm 0.0}}$ | $\mathbf{95.3_{\pm 0.0}}$ | $\mathbf{95.3_{\pm 0.0}}$ |
| | TabNet | $\mathbf{0.0_{\pm 0.0}}$ | $\mathbf{0.0_{\pm 0.0}}$ | $\mathbf{0.0_{\pm 0.0}}$ | $\mathbf{0.0_{\pm 0.0}}$ |
| WIDS | TabTr. | $61.9_{\pm 0.3}$ | $45.9_{\pm 0.3}$ | $22.6_{\pm 0.7}$ | $\mathbf{2.6_{\pm 0.5}}$ |
| | RLN | $69.8_{\pm 0.2}$ | $60.9_{\pm 0.2}$ | $41.5_{\pm 0.8}$ | $\mathbf{2.2_{\pm 0.4}}$ |
| | VIME | $61.4_{\pm 0.1}$ | $50.3_{\pm 0.2}$ | $27.8_{\pm 0.5}$ | $\mathbf{8.4_{\pm 0.5}}$ |
| | STG | $70.3_{\pm 0.1}$ | $63.8_{\pm 0.2}$ | $50.2_{\pm 0.1}$ | $\mathbf{7.8_{\pm 1.0}}$ |
| | TabNet | $29.4_{\pm 0.2}$ | $5.3_{\pm 0.4}$ | $\mathbf{1.9_{\pm 0.4}}$ | $5.2_{\pm 0.5}$ |

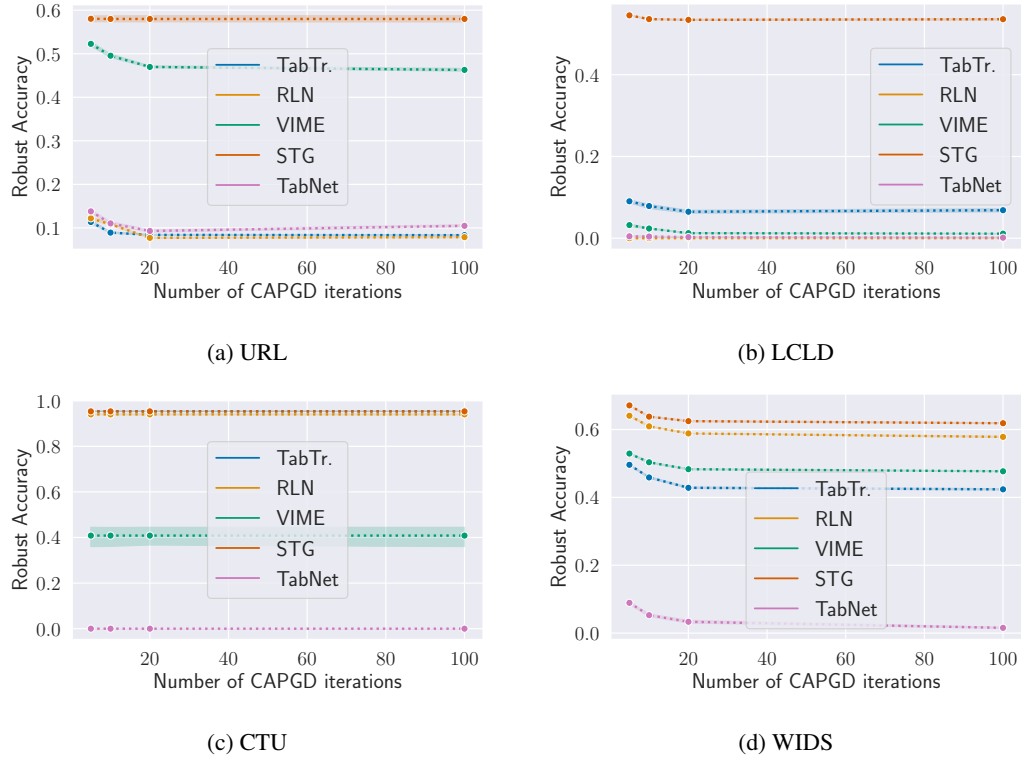

Figure 6: Robust accuracy with CAA with varying gradient attack iterations in CAPGD.

Table 10: Robust accuracy with CAA with varying gradient attack iterations in CAPGD. The lowest robust accuracy is in bold.

| Dataset | Model | # iterations CAPGD | | | |
| | | 5 | 10 | 20 | 100 |
| --- | --- | --- | --- | --- | --- |
| URL | TabTr. | $11.4_{\pm 0.5}$ | $8.9_{\pm 0.2}$ | $\mathbf{8.4_{\pm 0.1}}$ | $8.3_{\pm 0.1}$ |
| | RLN | $12.2_{\pm 0.4}$ | $10.8_{\pm 0.2}$ | $\mathbf{7.7_{\pm 0.1}}$ | $7.9_{\pm 0.2}$ |
| | VIME | $52.3_{\pm 0.6}$ | $49.5_{\pm 0.5}$ | $47.0_{\pm 0.2}$ | $46.3_{\pm 0.3}$ |
| | STG | $\mathbf{58.0_{\pm 0.8}}$ | $\mathbf{58.0_{\pm 0.8}}$ | $\mathbf{58.0_{\pm 0.8}}$ | $\mathbf{58.0_{\pm 0.8}}$ |
| | TabNet | $13.8_{\pm 0.4}$ | $11.0_{\pm 0.5}$ | $\mathbf{9.3_{\pm 0.3}}$ | $10.5_{\pm 0.3}$ |
| LCLD | TabTr. | $9.0_{\pm 0.5}$ | $7.9_{\pm 0.6}$ | $\mathbf{6.5_{\pm 0.4}}$ | $6.9_{\pm 0.5}$ |
| | RLN | $\mathbf{0.0_{\pm 0.0}}$ | $\mathbf{0.0_{\pm 0.0}}$ | $\mathbf{0.0_{\pm 0.0}}$ | $\mathbf{0.0_{\pm 0.0}}$ |
| | VIME | $3.2_{\pm 0.3}$ | $2.4_{\pm 0.1}$ | $1.2_{\pm 0.1}$ | $\mathbf{1.1_{\pm 0.0}}$ |
| | STG | $54.5_{\pm 0.1}$ | $53.6_{\pm 0.1}$ | $\mathbf{53.4_{\pm 0.2}}$ | $53.6_{\pm 0.2}$ |
| | TabNet | $0.5_{\pm 0.2}$ | $0.4_{\pm 0.1}$ | $0.3_{\pm 0.1}$ | $\mathbf{0.1_{\pm 0.1}}$ |
| CTU | TabTr. | $\mathbf{95.3_{\pm 0.0}}$ | $\mathbf{95.3_{\pm 0.0}}$ | $\mathbf{95.3_{\pm 0.0}}$ | $\mathbf{95.3_{\pm 0.0}}$ |
| | RLN | $\mathbf{94.0_{\pm 0.2}}$ | $\mathbf{94.0_{\pm 0.2}}$ | $\mathbf{94.0_{\pm 0.2}}$ | $\mathbf{94.0_{\pm 0.2}}$ |
| | VIME | $\mathbf{40.8_{\pm 4.7}}$ | $\mathbf{40.8_{\pm 4.7}}$ | $\mathbf{40.8_{\pm 4.7}}$ | $\mathbf{40.8_{\pm 4.7}}$ |
| | STG | $\mathbf{95.3_{\pm 0.0}}$ | $\mathbf{95.3_{\pm 0.0}}$ | $\mathbf{95.3_{\pm 0.0}}$ | $\mathbf{95.3_{\pm 0.0}}$ |
| | TabNet | $\mathbf{0.0_{\pm 0.0}}$ | $\mathbf{0.0_{\pm 0.0}}$ | $\mathbf{0.0_{\pm 0.0}}$ | $\mathbf{0.0_{\pm 0.0}}$ |
| WIDS | TabTr. | $49.6_{\pm 0.2}$ | $45.9_{\pm 0.3}$ | $\mathbf{42.8_{\pm 0.3}}$ | $42.4_{\pm 0.2}$ |
| | RLN | $64.1_{\pm 0.2}$ | $60.9_{\pm 0.2}$ | $58.8_{\pm 0.0}$ | $\mathbf{57.8_{\pm 0.2}}$ |
| | VIME | $52.9_{\pm 0.3}$ | $50.3_{\pm 0.2}$ | $48.3_{\pm 0.2}$ | $\mathbf{47.7_{\pm 0.1}}$ |
| | STG | $67.1_{\pm 0.1}$ | $63.8_{\pm 0.2}$ | $62.5_{\pm 0.2}$ | $\mathbf{61.8_{\pm 0.1}}$ |
| | TabNet | $8.9_{\pm 0.4}$ | $5.3_{\pm 0.4}$ | $3.3_{\pm 0.5}$ | $\mathbf{1.6_{\pm 0.2}}$ |

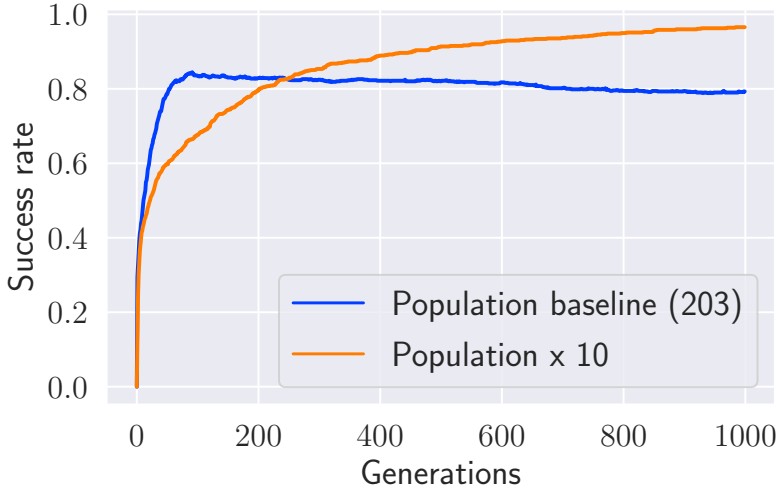

Figure 7: MOEVA success rate (LCLD - TabTransformer).

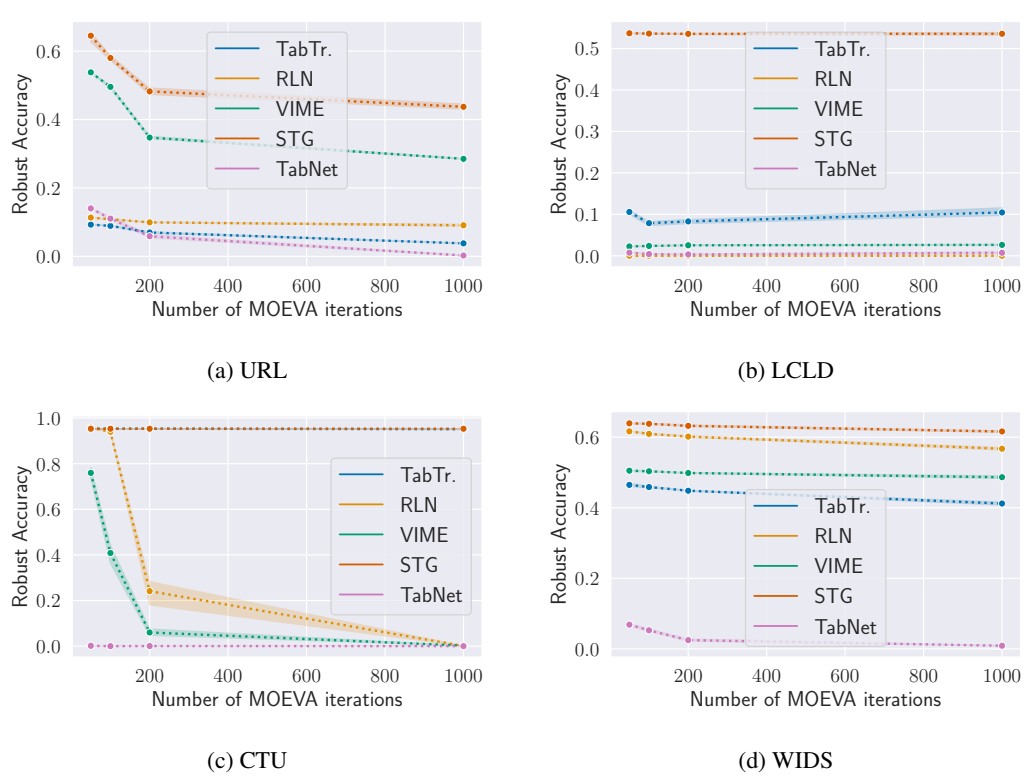

(a) URL

(b) LCLD

(c) CTU

(d) WIDS

Figure 8: Robust accuracy with CAA with varying search attack iterations in MOEVA.

Table 11: Robust accuracy with CAA with varying search attack iterations in MOEVA. The lowest robust accuracy is in bold.

| Dataset | Model | # iterations MOEVA | | | |
| | | 50 | 100 | 200 | 1000 |
|---|---|---|---|---|---|
| URL | TabTr. | $9.3_{\pm0.0}$ | $8.9_{\pm0.2}$ | $7.0_{\pm0.2}$ | $\mathbf{3.8_{\pm0.1}}$ |
| | RLN | $11.3_{\pm0.2}$ | $10.8_{\pm0.2}$ | $9.9_{\pm0.1}$ | $\mathbf{9.1_{\pm0.4}}$ |
| | VIME | $53.8_{\pm0.2}$ | $49.5_{\pm0.5}$ | $34.7_{\pm0.4}$ | $\mathbf{28.5_{\pm0.2}}$ |
| | STG | $64.5_{\pm1.7}$ | $58.0_{\pm0.8}$ | $48.2_{\pm1.0}$ | $\mathbf{43.7_{\pm0.8}}$ |
| | TabNet | $14.0_{\pm0.2}$ | $11.0_{\pm0.5}$ | $5.9_{\pm0.6}$ | $\mathbf{0.3_{\pm0.1}}$ |
| LCLD | TabTr. | $10.6_{\pm0.6}$ | $\mathbf{7.9_{\pm0.6}}$ | $8.3_{\pm0.5}$ | $10.5_{\pm1.1}$ |
| | RLN | $\mathbf{0.0_{\pm0.0}}$ | $\mathbf{0.0_{\pm0.0}}$ | $\mathbf{0.0_{\pm0.0}}$ | $\mathbf{0.0_{\pm0.0}}$ |
| | VIME | $\mathbf{2.3_{\pm0.1}}$ | $2.4_{\pm0.1}$ | $2.5_{\pm0.1}$ | $2.6_{\pm0.1}$ |
| | STG | $53.7_{\pm0.1}$ | $53.6_{\pm0.1}$ | $\mathbf{53.5_{\pm0.2}}$ | $53.6_{\pm0.4}$ |
| | TabNet | $0.8_{\pm0.2}$ | $0.4_{\pm0.1}$ | $\mathbf{0.3_{\pm0.1}}$ | $0.7_{\pm0.3}$ |
| CTU | TabTr. | $95.3_{\pm0.0}$ | $95.3_{\pm0.0}$ | $95.3_{\pm0.0}$ | $\mathbf{95.1_{\pm0.2}}$ |
| | RLN | $95.8_{\pm0.4}$ | $94.0_{\pm0.2}$ | $24.1_{\pm6.1}$ | $\mathbf{0.0_{\pm0.1}}$ |
| | VIME | $76.0_{\pm2.7}$ | $40.8_{\pm4.7}$ | $6.0_{\pm1.5}$ | $\mathbf{0.2_{\pm0.0}}$ |
| | STG | $\mathbf{95.3_{\pm0.0}}$ | $\mathbf{95.3_{\pm0.0}}$ | $\mathbf{95.3_{\pm0.0}}$ | $\mathbf{95.3_{\pm0.0}}$ |
| | TabNet | $0.1_{\pm0.1}$ | $\mathbf{0.0_{\pm0.0}}$ | $\mathbf{0.0_{\pm0.1}}$ | $\mathbf{0.0_{\pm0.0}}$ |
| WIDS | TabTr. | $46.5_{\pm0.5}$ | $45.9_{\pm0.3}$ | $44.8_{\pm0.2}$ | $\mathbf{41.2_{\pm0.5}}$ |
| | RLN | $61.7_{\pm0.2}$ | $60.9_{\pm0.2}$ | $60.2_{\pm0.3}$ | $\mathbf{56.7_{\pm0.4}}$ |
| | VIME | $50.5_{\pm0.2}$ | $50.3_{\pm0.2}$ | $49.9_{\pm0.2}$ | $\mathbf{48.7_{\pm0.4}}$ |
| | STG | $63.9_{\pm0.2}$ | $63.8_{\pm0.2}$ | $63.2_{\pm0.3}$ | $\mathbf{61.6_{\pm0.3}}$ |
| | TabNet | $6.9_{\pm0.4}$ | $5.3_{\pm0.4}$ | $2.5_{\pm0.4}$ | $\mathbf{0.9_{\pm0.1}}$ |

## B.3 Constraints complexity

In this section, we study the impact of the constraints' complexity on CAA's effectiveness.

CAPGD and MOEVA fail to generate adversarial examples on CTU for 2 out of 5 models. CTU dataset has a large number of constraints compared to the other datasets, and some are particularly challenging. We argue that some of these constraints hinder gradient attacks and are harder to optimize. To confirm our hypothesis and provide additional insights, we split the constraints of CTU based on their complexity. We consider two aspects of constraint complexity: the number of constraints to satisfy and the number of features involved in a single constraint.

We split the constraints of CTU as follows:

- **CG0** One constraint involving 90 features in the form of $\sum F_i = \sum F_j$ where both sums represent the total number of sent packets.

- **CG1** One constraint involving 90 features in the form of $\sum F_i = \sum F_j$ where both sums represent the total number of received packets.

- **CG2** 34 constraints in the form of $BYTE/PACKETS \leq 1500$, to model the fact that each packet contains at most 1500 bytes.

- **CG3** 324 constraints in the form of $A \leq B$ where A and B are statistical properties (min, max, sum) for each port, and direction.

First, we ran an ablation study, where we ignored one bucket of constraint at a time. Next, we studied the success rate when we considered each bucket separately. Finally, we reported the impact of the number of constraints to optimize from CG3, the largest bucket.

The results in Table 12 show that for gradient attacks, removing one type of constraint is not enough to improve the success rate. Constraints across multiple remaining categories are not satisfied. The individual bucket study confirms that only when considering constraints of type CG2 alone, CAPGD improves its success rate (in VIME and TabNet). When only considering CG3 constraints, reducing the number of constraints improves the success rate (by reducing robust accuracy from 95.3% when considering 100% of CG3 constraints to 84.5% and 43.0% respectively when considering 50% and 10% of the constraints).

Table 12: Robust accuracy with subset of constraints. $\Omega$ is the complete set of constraints. CGX denotes the constraint group X. For CG2 and CG3, we evaluate with the entire group and on 10%, 25%, 50% selected randomly and averaged on 5 seeds.

| | | CAPGD | | | | | CAA | | | | |
|---|---|---|---|---|---|---|---|---|---|---|---|
| | Group | RLN | STG | TabNet | TabTr. | VIME | RLN | STG | TabNet | TabTr. | VIME |
| | $\Omega$ | 97.8 | 95.3 | 96.1 | 95.3 | 95.1 | 94.0 | 95.3 | 0.0 | 95.3 | 40.8 |
| Ablation | $\Omega \setminus CG0$ | 97.8 | 95.3 | 96.1 | 95.3 | 95.1 | 93.9 | 95.3 | 0.0 | 95.3 | 21.0 |
| | $\Omega \setminus CG1$ | 97.8 | 95.3 | 96.1 | 95.3 | 95.1 | 94.1 | 95.3 | 0.0 | 95.3 | 37.1 |
| | $\Omega \setminus CG2$ | 97.8 | 95.3 | 96.1 | 95.3 | 95.1 | 93.9 | 95.3 | 0.1 | 95.3 | 40.8 |
| | $\Omega \setminus CG3$ | 97.8 | 95.3 | 96.1 | 95.3 | 95.1 | 89.3 | 95.3 | 0.0 | 95.3 | 2.4 |
| Components | CG0 | 97.8 | 95.3 | 96.1 | 95.3 | 95.1 | 91.6 | 95.3 | 0.0 | 95.2 | 2.8 |
| | CG1 | 97.8 | 95.3 | 96.1 | 95.3 | 95.1 | 91.1 | 95.3 | 0.0 | 95.2 | 1.9 |
| | CG2 | 75.3 | 95.3 | 31.4 | 94.3 | 0.0 | 72.0 | 95.3 | 0.0 | 94.3 | 0.0 |
| | CG3 | 97.2 | 95.3 | 95.3 | 95.3 | 95.1 | 92.7 | 95.3 | 0.0 | 95.3 | 19.0 |
| Percentage CG3 | 10% | 85.2 | 95.3 | 43.0 | 95.1 | 11.3 | 80.8 | 95.3 | 0.0 | 94.9 | 0.6 |
| | 25% | 93.4 | 95.3 | 59.9 | 95.3 | 37.8 | 87.1 | 95.3 | 0.0 | 95.3 | 2.5 |
| | 50% | 94.9 | 95.3 | 84.5 | 95.3 | 93.2 | 88.6 | 95.3 | 0.0 | 95.3 | 8.3 |

## B.4  Additional defenses

Table 13: CAA performances against Madry Adversarially Trained model, Adversarial Training + Cutmix and Adversarial Training + CT-GAN.

| Dataset | Training | Architecture | | | | |
|---|---|---|---|---|---|---|
| | | RLN | STG | TabNet | TabTr | VIME |
| URL | Adversarial Training | 56.2 | 90.0 | 91.8 | 56.7 | 69.8 |
| | Adv. Tr. + Cutmix | 60.8 | 42.7 | 89.7 | 40.3 | 68.6 |
| | Adv. Tr. + CT-GAN | 62.5 | 79.8 | 89.9 | 66.0 | 66.9 |
| LCLD | Adversarial Training | 63.0 | 12.1 | 0.0 | 70.3 | 10.4 |
| | Adv. Tr. + Cutmix | 47.0 | 36.2 | 0.0 | 71.0 | 52.9 |
| | Adv. Tr. + CT-GAN | 54.3 | 81.2 | 0.0 | 78.5 | 76.8 |
| CTU | Adversarial Training | 97.1 | 95.1 | 0.2 | 95.3 | 94.0 |
| | Adv. Tr. + Cutmix | 95.3 | 94.5 | 0.0 | 95.3 | 94.3 |
| | Adv. Tr. + CT-GAN | 96.7 | 96.0 | 100.0 | 94.4 | 100.0 |
| WIDS | Adversarial Training | 66.6 | 45.2 | 58.4 | 65.1 | 52.1 |
| | Adv. Tr. + Cutmix | 59.9 | 41.2 | 37.4 | 50.8 | 43.5 |
| | Adv. Tr. + CT-GAN | 100.0 | 73.8 | 100.0 | 68.1 | 100.0 |

In addition to the application of adversarial training alone, we evaluate the robustness of the models when training with data augmentation. [32] showed that combining data augmentation with adversarial training can increase the robustness of models. We also observe in RobustBench [12], that top-performing models are trained with data augmentation. We consider two data augmentation techniques: Cutmix [45] and CT-GAN [42]. To train our model, at each batch, we use an equal proportion of clean, adversarial examples, data augmentation examples, and adversarial examples generated on top of data augmentation examples.

We evaluated CAA on these two defenses and report in Table 13 the robust accuracy of these defenses, compared to the robustness of the vanilla Madry AT on all our datasets and models. The results are the average over 5 seeds run to ensure reliable evaluation. Our experiments show that these new defenses can significantly improve the robustness of the models to CAA, but that our new attack remains effective for URL and LCLD datasets across all architectures, and for WIDS on TabTransformer and STG architectures.

Table 14: Tree-based model robust accuracy in direct and transferability scenario (minimum robust accuracy over 5 neural networks).

| Dataset | Model | Clean | Direct | Transferability |
|---------|-------|-------|--------|-----------------|
| URL | Random Forest | 96.2 | 52.7 | 72.4 |
|     | XGBoost | 97.4 | 27.3 | 46.7 |
| LCLD | Random Forest | 64.3 | 22.3 | 5.3 |
|      | XGBoost | 68.3 | 9.1 | 9.4 |
| CTU | Random Forest | 95.6 | 92.2 | 95.0 |
|     | XGBoost | 97.3 | 76.3 | 97.1 |
| WIDS | Random Forest | 52.2 | 5.2 | 14.1 |
|      | XGBoost | 80.4 | 38.0 | 60.7 |

## B.5 Generalization to shallow models

Shallow models and in particular tree-based models such as Random Forest (RF) XGBoost remain among the best models for tabular data on average [5]. We evaluate the robustness of these models against CAA in two settings: (1) direct attack where CAA (using its search component MOEVA) attacks directly the RF and XGBOOST models, and (2) transfer attacks, where we craft the examples on our deep learning (DL) models and evaluate them in the RF and XGBOOST models. Table **??** shows that (1) DL models of our study achieve comparable clean performance to the shallow models, (2) both RF and XGBOOST models are vulnerable to direct CAA attacks (down to 9.1% of robust accuracy on LCLD XGBoost), and (3) CAA attacks on DNN transfer to RF (down to 5.3% robust accuracy) and XGBoost (down to 9.4% robust accuracy) models.

