# OpenReview forum: "Constrained Adaptive Attack: Effective Adversarial Attack Against Deep Neural Networks for Tabular Data"
_NeurIPS.cc/2024/Conference — NeurIPS 2024 spotlight_

### Official Review · Reviewer_aMTd · 2024-06-22

**Soundness:** 3
**Presentation:** 3
**Contribution:** 3
**Rating:** 5
**Confidence:** 4

**Summary:**

This paper aims to study the problem of finding adversarial examples for tabular datasets. Different from attacking image models or text models, attacking tabular models requires finding adversarial examples which are legal, which do not violate the relation between features. Moreover, it also requires to tackle both numerical and categorical features.

**Strengths:**

1. The studied problem is important and under less investigation compared to adversarial attacks in CV and NLP.
2. The paper is well written and easy to follow.
3. The experiment analysis is convincing and sufficient.

**Weaknesses:**

However, I still have concerns on this paper. Specifically, many terms / techniques used in the method are not originally proposed by this paper. For example,
1. In terms of the main objective (as discussed in Section 3.1), one key step to let adversarial examples satisfy the feature constraints (in Eq.1 and Eq.2) is to translate into a differentiable function. However, this is proposed by the previous work Simonetto et al, 2021.
2. In terms of the solving algorithm (as discussed in Section 4.1), the author propose CAPGD, which seems a fine-tuning / lightweight modification of the previous method CPGD. This also limit the technical contribution of this paper.
3. Finally, the author claims trying both CAPGD (proposed in this paper) and MOEVA (previous work) and BF* (previous work) can improve the attack successful rate.

Overall, I do not hesitate about the effectiveness of combining these strategies for solving the attack problem. Instead, I am concerned of the originality and the significance of the work. Besides, although the authors discussed the reasons on why differentiable models are considered in this paper, I still believe that undifferentiable models like XGBoost is one important branch of models for solving the tabular data classification tasks. I encourage the authors also investigate possible solutions for undifferentiable models.

**Questions:**

Plz see the weakness.

**Limitations:**

I didn't see such a discussion.

---

> ### Author Rebuttal · Authors · 2024-08-07
>
> Thank you for your review. We appreciate that you acknowledge the importance of the problem and the quality and persuasiveness of our analysis. We will address below your doubts about the originality and significance of our work.
>
> **W1 - Existing differentiable function:**
>
> We acknowledge that integrating the constraints as a differentiable penalty function is not novel in this work and was proposed by CPGD [33]. Nevertheless, the results in [33] (confirmed in Table 2 of our paper) indicate that leveraging the constraints penalty function is not sufficient to achieve high effectiveness with gradient-based attacks. An attack that only uses a differentiable penalty such as CPGD remains ineffective in the challenging datasets we address. Thus, the need for the novel mechanisms we introduce in CAPGD and CAA.
>
> **W2 - CAPGD novelty:**
>
> We understand your concern regarding the similarities between CPGD and CAPGD, being both gradient attacks where the constraints are included as penalty functions. However, we respectfully disagree for the following reasons:
>
> (1) Compared to the previous attack, CPGD, our new attack CAPGD introduces 4 novel mechanisms to increase the effectiveness of the attack: the repair operator, the adaptive step, the momentum, and multiple initialization.
> We demonstrate the effectiveness and complementarity of each component.
> For instance, removing the constraints repair operator, a novelty of this paper, reduces the effectiveness of the attack by up to 24.1 robust accuracy points.
> CAPGD significantly increases the effectiveness of gradient-based attacks on 3/4 datasets, while preserving the efficiency advantage of gradient-based attacks w.r.t. search attacks.
>
> (2) Designing CAPGD is not intuitive. The literature on adversarial attacks proposed many mechanisms to improve their effectiveness, including random sampling, reinitialization, adaptive steps, and revert to best, and the first challenge was to identify which mechanisms are relevant to our use case, namely which mechanisms are compatible and beneficial to the constraints satisfaction objective. In particular, we found out that combining all the mechanisms in the literature was not optimal, and we achieved the best performances without reverting to the example with the higher loss when the step size changes. We provided in Appendix B.1 a study of the components of CAPGD and their complementarity.
>
> We hope this clarifies our argument on why CAPGD is not a straightforward improvement of CPGD. We are open to further discussion and would be happy to elaborate on any points if needed.
>
>
> **W3 - Applying a combination of attacks:**
>
> We appreciate your concern regarding the design of our meta-attack CAA.  Nevertheless, we respectfully believe that its design represents a significant contribution for two reasons:
>
> (1) By applying a combination of complementary and strong attacks, we aim to become the standard for evaluating the robustness of models to adversarial attacks in Tabular Machine Learning.
>
> Identifying both "complementary" and "strong" attacks for a meta-attack is not straightforward. We presented in Table 1 of our manuscript 10 attacks proposed for Tabular Machine Learning. Some are gradient-based and others are search-based. We first identified which attacks natively support all the constraints, and which could eventually be extended. We ended with 2 gradient attacks CPGD (native support) and LowProFool (extension needed), and two search attacks BF* and MOEVA. Next, we demonstrated that our new attack CAPGD subsumes CPGD and LowProfool and that MOEVA subsumes BF*. Then we needed to validate the complementarity of CAPGD and MOEVA (Figure 2 of our manuscript). Finally, we confirmed with an extensive evaluation (Table 3 of our manuscript) that our new CAA attack combining MOEVA and CAPGD still preserves the benefits of each (efficiency of CAPGD and effectiveness of MOEVA). Each step required formal analysis, extensive engineering, and exhaustive experiments.
>
> (2) Meta-attacks, that combine existing attacks are an active line of research. In Computer Vision, AutoAttack by Croce et al.[12] is a meta attack combining existing attacks (APGD+FAB) to benefit from the complementarity of search and gradient attacks. The attack has revolutionized the research of robustness assessment in computer vision and led to a significant improvement of novel defense mechanisms.
>
> We strongly believe that CAA will be as impactful for Tabular Machine Learning as AutoAttack, especially as it was carefully designed to achieve both efficiency for simple cases and effectiveness for challenging cases and has very limited hyper-parameters.
>
> Thank you once again for your constructive feedback, we will update the final manuscript according to your feedback to better showcase the significance of our meta-attack.
>
> **W4 - Undifferentiable models:**
>
> We agree that tree-based models such as XGBoost are in many settings outperforming Deep Neural Network (DNN) architectures. DNN architectures are catching up, in particular for large datasets as demonstrated in [5], hence the need to evaluate their robustness.
>
> However, we also argue that CAA is relevant to any model (including tree-based models) with two settings:
>
> - in transferability, by generating adversarial examples on a surrogate model and evaluating their success rate on the target (tree-based) model,
> - by applying CAA (through the search-based component MOEVA that is model agnostic).
>
> In the common author response (C3), we provide an empirical study for both settings and show that CAA remains effective against random forest and XGBOOST models.
>
> We will also update the appendix in the final version of the paper with a discussion on undifferentiable models.

---

> > ### Comment · Reviewer_aMTd · 2024-08-11
> >
> > We thank the reviewer's response. I agree with the reviewer that this method is potential to be served as a good baseline strong attack in the related literature, thanks to the authors' effort on trying different strategies for performance improvement. It is also insightful to see the transferability of attacking between different types of models. Thus, I increase the rating to 5.

---

> > > ### Author Response · Authors · 2024-08-13
> > >
> > > We thank you for your response and positive feedback on our extended analysis and new results. We appreciate that you have increased your score to 5 and are happy to provide any additional insight or answer any interrogation you may have to fully satisfy your requirements for a clear acceptance of our work.

---

### Official Review · Reviewer_5k6D · 2024-07-03

**Soundness:** 4
**Presentation:** 3
**Contribution:** 3
**Rating:** 7
**Confidence:** 4

**Summary:**

This paper introduces two novel adversarial attacks, specifically designed to target the evasion of deep neural networks (DNNs) in classification tasks involving tabular data satisfying real-world constraints. The first attack, CAPGD, is a gradient-based method that enhances the constrained PGD (CPGD) attack proposed in [33] by incorporating improvements from the APGD definition [12], such as the addition of momentum and an optimized step size selection. The second attack, CAA, combines the efficient but less effective CAPGD with another search-based attack, MOEVA, which is more effective but less efficient regarding the computational time required to perform the attack.
The extensive experimental evaluation on four tabular datasets and five DNNs for tabular data demonstrates that CAPGD significantly improves the success rate of generating successful adversarial examples compared to CPGD. Furthermore, CAA achieves a reasonable trade-off between efficacy and computational efficiency, offering higher attack efficacy than CAPGD and requiring less computational time than MOEVA.

**Strengths:**

I find this paper particularly compelling as it proposes a new evasion attack against deep learning models for tabular data with real constraints that clearly outperforms the other state-of-the-art proposals. The paper demonstrates two key strengths:

- Despite the two proposed attacks are improvements and combinations of existing approaches to generate realistic attacks, they are based on reasonable and novel observations. In particular, the design of CAA is motivated by the evidence that CAPGD and MOEVA are complementary. These attacks significantly outperform the state-of-the-art CPGD [33].

- The experimental analysis is comprehensive and convincing. The experiments are performed on 5 different deep learning models employed to classify tabular data on 4 different datasets with variable number of constraints. Both the accuracy and efficiency of the attacks are evaluated. Furthermore, additional results are shown varying the maximum perturbation and the value of important parameters of the attacks are shown. Finally, an ablation study shows the impact of each improvement introduced in CAPGD.

**Weaknesses:**

Even though the proposal is good, I think that some weaknesses about the presentation of the proposal and the analysis of the results need to be addressed in the final version of the paper:

**The description of CAPGD should be improved**: the description of CAPGD can be enhanced by better explaining the role of the repair operator. Specifically, the authors mention that constraints of the form $f = \psi$ are enforced at every iteration by the repair operator. However, it is not clear whether applying this operator at every iteration guarantees the generation of an adversarial example that satisfies all dataset constraints and the maximum perturbation constraint by the end of the execution. This is a crucial property, as the attack cannot be tested if it does not satisfy the constraints, potentially wasting the time required to generate the attack.

**The depth of the analysis of the results should be improved**: the depth of the analysis of the results can be improved by discussing particular results to understand their causes and how to mitigate these cases. For instance, CAPGD and MOEVA fail to generate adversarial examples on the CTU dataset for 2 out of 5 models (Table 3 and results in the Appendix). This failure may be due to the high number of constraints (360) considered for this dataset, since the two attacks are more effective on the other datasets for which at most 30 constraints are considered. It would be interesting to evaluate the efficacy of the two attacks by varying the number of constraints considered for the CTU dataset, to better highlight the limitations of the proposed attacks.
Additionally, CAA generates less effective attacks with higher epsilon values (Figure 5b), and MOEVA generates less effective attacks with a higher number of iterations (Figure 7b). Even though these results are sporadic, they are counterintuitive, as the search space for adversarial examples increases with higher epsilon and more mutation rounds. A detailed discussion of these phenomena would provide valuable insights.

**The limitations section is missing**: the paper should include a limitations section summarizing the limitations of the presented attack algorithms and their evaluation. For example, the evaluation is performed only against an $L_2$ attacker, whereas other norm-based or synthetic attackers could have been considered. The attacks may not work well against specific datasets and models when $\epsilon$ is small. While these limitations are reported in the paper, they are scattered throughout. A dedicated section would clarify these points.

**Typos and inconsistencies**: there are some typos and inconsistencies in the notation, such as:
- $R_\Omega$ in Eq. 3 is not defined when the equation is presented.
- $x^{(k+1)}$ -> $x^{(k-1)}$ in Eq. 7.
- The classifier is represented by $H$ in Algorithm A.2, but $h$ is used in the main body of the paper. Additionally, $X$ appears twice in the function in line 3 of Algorithm A.2.

Correcting these typos and inconsistencies will improve the clarity and readability of the paper.

Finally, another weakness is more generic:

**The proposed attacks are really specific**: the paper is proposing two effective algorithms to forge realistic evasion attacks against deep learning models used for classifying tabular data, an under-explored setting. I think that the contribution is appreciable. However, the chosen setting may limit the impact of the results, since, at the best of my knowledge, tree-based models continue to be the state-of-the-art for classification tasks on tabular data [a].

[a] Ravid Shwartz-Ziv and Amitai Armon, Tabular Data: Deep Learning is Not All You Need, Information Fusion Volume 81, May 2022, Pages 84-90

**Updates after the authors' response**
The authors have provided a satisfying response addressing all the concerns that I have raised as weaknesses. The authors are willing to introduce the findings provided in the response in the next version of their paper, as far as I understood. Moreover, they will also include a dedicated limitations section and correct the typos.

**Questions:**

- Is CAPGD guaranteed to generate an attack that satisfies the dataset and perturbation constraints at the end of execution?

- CAPGD and MOEVA may be unable to generate adversarial examples in specific settings (see results on two models on the CTU dataset in Table 3). What is the reason? Could it be due to the high number of constraints of the dataset? How does the success rate of the two attacks vary considering different numbers and types of constraints (linear and nonlinear)?

- Why may CAA and MOEVA show a smaller success rate when considering higher epsilon values and a higher number of rounds (see Figures 5b and 7b)?

**Limitations:**

The authors should sum up the limitations of their proposals in a specific section in the paper. The negative societal impact is discussed by the authors who claim that their work may give birth to new and stronger defenses. Finally, the experimental and implementation details have been extensively documented.

---

> ### Author Rebuttal · Authors · 2024-08-07
>
> Thank you for your support and your insightful feedback. We appreciate your comments towards improving the quality of the paper. We clarify and answer your concerns below.
>
> **W1/Q1 - Explaining the role of the repair operator. Is CAPGD guaranteed to generate an attack that satisfies the dataset and perturbation constraints at the end of execution?**
>
> The repair operator's role is to ensure equality constraints are satisfied during optimization.
> While equality constraints are included in the penalty function, optimization alone does not achieve exact equality of feature values.
> The repair operator addresses this by setting the value of the left-hand side of an equation to match the evaluation of the right-hand side in each iteration.
> It maintains other dataset constraints such as bounds, mutability, and feature types but does not ensure other relational constraints are met.
> The operator can violate maximum perturbation constraints, yet at each iteration, the perturbation is corrected back within the allowed maximum.
> This approach has been shown to improve the success rate of CAPGD, as demonstrated by our ablation study in Table 7.
>
> **W2/Q2 - Attacks fail to generate adversarial examples on CTU for 2/5 models. Is it the high number of constraints? Impact of different numbers and types of constraints ?**
>
> Thank you for raising this comment. Indeed, CTU dataset exhibits a large number of constraints compared to the other datasets, and some are particularly challenging. We show in the common answer (C1) that these constraints are harder to optimize.
>
> **W2/Q3 - CAA epsilon values - Why may CAA show a smaller success rate when considering higher epsilon values?**
>
> Thank you for pointing out the pattern in Fig5b. Indeed, CAA's performance can decrease with higher budgets. We provide below a complementary analysis of this behavior.
>
> We investigated the case of augmenting the EPS budget of CAA on LCLD dataset with STG model.
> We found that when augmenting the EPS budget from 1 to 5, the success rate of CAPGD drops from 27.1% to 0.4% and is not entirely balanced by MOEVA's improvement from 4.3% to 12.4%.
>
> The drop in CAPGD performance is caused in almost all cases by the violation of boundary constraint after the repair operator when the repair operator fixes the constraints of the type A = B / C.
>
> Each of these features is defined in the dataset with their respective maximum and minimum values. Given Max(B) and Min(C), the repair operator will lead to Max(A) = Max(B) / Min( C), however in the definition of the dataset proposed by Simonetto et. al. [33], Max (A) is lower than Max(B) / Min( C). Hence our repair operator contradicts the boundary constraint definitions. This phenomenon appears only for large perturbations. This violation should be solved by fixing the dataset's definition of the boundary constraints to take into account the feature relationships.
>
> We have included in the limitation section of our paper (in the final version, and in the author's response above C3) a discussion on the quality of the constrained datasets available and the coherence between their boundary constraints and their feature constraints.
>
> **W2/Q3 - MOEVA Iterations - Why may MOEVA show a smaller success rate with a higher number of generations?**
>
> MOEVA is a multi-objective genetic algorithm. An inherent problem of multi-objective optimization is the trade-off between the objectives. If all solutions in the population are on the Pareto front, the algorithm must decide which solutions to discard for the next iteration, potentially discarding a valid adversarial example in our case.
>
> Figure 1 in the Author's response PDF shows the evolution of the success rate of MOEVA with the number of iterations in the same settings as in Figure 7b for TabTransformer.
> We find that the success rate reaches a maximum at 100 iterations.
> We argue that valid adversarial examples were discarded when the search continued to 1000 iterations.
> To confirm our hypothesis, we run the same experiment with a 10 times larger search population, such that more solutions are preserved at each iteration.
> We observe that in this setting, MOEVA converges slower (due to less selection pressure) but the success rate strictly increases with the number of generations.
> Increasing the population size also increases the execution time (by 3.4x in this case), due to the selection operator overhead.
>
> Our approach CAA aims at minimizing the memory and computation overheads while maximizing the success rate, and CAA can be tuned to lead to lower robust accuracy with more iterations if the search space is expanded (for example with larger populations). We thank you for this remark and we have introduced a discussion on the impact of the population size in the appendix.
>
> **W3 - Limitation sections:**
>
> Thank you for this suggestion. We added a limitation section in the common author response (C4), and to the updated paper for the final version.
>
> **W4 - Typos:**
>
> Thank you for pointing this out, we have corrected them for the final version.
>
> **W5 - The proposed attacks are specific:**
>
> We agree that tree-based models are in many settings outperforming Deep Neural Network (DNN) architectures. DNN architectures are catching up, in particular for large datasets as demonstrated in [A], hence the need to evaluate their robustness.
>
> However, we also argue that CAA is relevant to any model (including tree-based models) with two settings:
>
> - in transferability, by generating adversarial examples on a surrogate model and evaluating their success rate on the target (tree-based) model,
> - by applying CAA (through the search-based component MOEV, which is model agnostic).
>
> In the common author response, we provide an empirical study for both settings and show that CAA remains effective against random forest and XGBOOST models.
>
> [A] Borisov et al. "Deep neural networks and tabular data: A survey. 2021.

---

> ### Comment · Reviewer_5k6D · 2024-08-08
> **Compliments to the authors for their high-quality response.**
>
> I sincerely thank the authors for their wide and deep response. Their points sound reasonable and clarify the open points that I raised in the review. I hope that the authors will discuss their new results in the next version of their paper.
>
> I have only an observation about the results in Table 3 of the attached PDF. It seems to me that attacking STG is difficult independently of the considered constraints since its robust accuracy is always 95.3%. Understanding if the model is really robust or if a better attack is needed could be an interesting future work.
>
> I will modify my review to acknowledge your response and improve my score, given the depth of your analysis and dedication you have shown in your response.

---

> > ### Author Response · Authors · 2024-08-13
> >
> > We thank you for your praise of our rebuttal and we are happy to see you increase your score following our answers.
> > To answer your observation on Table 3 on the author's response PDF. We confirm that STG architecture is the hardest to attack both with gradient and search attacks. We show consistent robustness of STG in Table 3 of our main submission across all the datasets.
> >
> > STG [A] is a novel architecture that implements an embedded nonlinear feature selection method by introducing the stochastic gates to the input layer (the feature space) of a neural network. The randomness introduced on the fly to select the features for training and inference significantly hinders evasion attacks, both gradient and search attacks.
> >
> > There is a similar phenomenon in the benchmark by Croce et al. for computer vision (Robustbench). In their evaluation of AutoAttack, they clearly "rule out classifiers which have (1) zero gradients with respect to the input, (2) randomized classifiers, and (3) classifiers that use an optimization loop at inference time" [B].
> > Contrary to their benchmark, we decided to cover one architecture per family of mechanisms from the Tabular ML literature, including stochastic mechanisms (STG belongs to categories 2 and 3 discarded in Robustbench).
> >
> > In Table 3 of our main paper, while we demonstrate that STG is the most robust, our study uncovers some cases where STG robust accuracy could still be significantly decreased with CAA (LCLD, URL), while other datasets will be challenging targets for future research (CTU).
> >
> > We appreciate your feedback on this pattern and the discussion it raises. In the appendix, we described the mechanisms of each architecture but did not connect the mechanisms with the observed robustness. Following your insight, we will also update this section to explain how each mechanism could impact the robustness (a complete verification of each mechanism impact would be a natural follow-up for dedicated papers).
> >
> > Thank you again for this discussion and the points you raised. They have significantly improved our final version.
> >
> > [A] Yamada, Yutaro, et al. "Feature selection using stochastic gates." ICML, 2020.
> >
> > [B] Croce et al. "Robustbench: a standardized adversarial robustness benchmark."(NeurIPS, 2021).

---

> > > ### Comment · Reviewer_5k6D · 2024-08-13
> > > **Official Comment by Reviewer 5k6D**
> > >
> > > I appreciate your investigation, which provides an intuition about the robustness of STG. Adding these observations to the paper will certainly strengthen your contribution. Thanks again; I will continue the discussion with the reviewers.

---

### Official Review · Reviewer_JEHw · 2024-07-09

**Soundness:** 2
**Presentation:** 3
**Contribution:** 2
**Rating:** 6
**Confidence:** 4

**Summary:**

The paper proposes two adversarial attack methods targeting deep learning models for tabular data. The two methods are: CAPGD (Constrained Adaptive Projected Gradient Descent) and CAA (Constrained Adaptive Attack). CAPGD is a modification based on constrained PGD with step size adjusting, repair operator, additional random initialization, and momentum. CAA is a combination of CAPGD and MOEVA (Multi-Objective Evolutionary Adversarial Attack), which is a search-based attack method. The two methods are combined by iteratively applying the two with CAPGD first. The authors demonstrate the effectiveness of these attacks across five architectures and four datasets.

**Strengths:**

Strengths:

1. Clarity: The motivation and rationale behind the proposed algorithms are clearly presented in a logical order.

2. The paper provides an extensive empirical evaluation of the proposed attacks across multiple datasets and architectures, showcasing their superiority in terms of effectiveness and computational efficiency.

**Weaknesses:**

Weakness:

1.	Lack of novelty and contribution: CAPGD is modified base don CPGD with a series of commonly used optimization techniques. CAA is a combination of CAPGD and MOEVA, which was proposed by a previous work.
2.	The implementation of CAA, involving the combination of CAPGD and MOEVA, might be complex and resource-intensive, which could limit its practical applicability in some settings.
3.	Lack of evaluation against defense mechanisms: Section 5.4 briefly discussed potential defense mechanisms, Madry’s adversarial training, which was proposed six years ago. There have been many works proposed since then. It is not fair to say that it is the only reliable defense against evasion attacks.

Minor:

•	On page 1, line 30, “This raises anew the need to study …”. Maybe it should be “This raises a new need to study …”?

**Questions:**

NA, see above.

**Limitations:**

No discussion found for limitation.

---

> ### Author Rebuttal · Authors · 2024-08-07
>
> Thank you for your feedback.
> We appreciate the opportunity to clarify and address any misunderstandings.
> We will address each of your points one by one and welcome further discussion on these issues.
>
> **W1 - Lack of novelty and contribution. CAPGD is modified based on CPGD. CAA is a combination of CAPGD and MOEVA:**
>
> We agree that the core intuitions behind the improvements of CAPGD and CAA  are elegant and not particularly complex, but we would like to point out that:
>
> (1) the literature on adversarial attacks proposed many mechanisms to improve their effectiveness, including random sampling, reinitialization, adaptive steps, and revert to best, and the first challenge was to identify which mechanisms are relevant to our use case, namely which mechanisms are compatible and beneficial to the constraints satisfaction objective. In particular, we found out that combining all the mechanisms in the literature was not optimal, and we achieved the best performances without reverting to the example with the higher loss when the step size changes. We provided in Appendix B.1 a study of the components of CAPGD and their complementarity. In addition, we also proposed new iterative repair mechanisms that were not explored in previous work and demonstrated their effectiveness.
> Hence, CAPGD is not a straightforward improvement of CPGD.
>
> (2) meta attacks, that combine existing attacks are also an active line of research. In Computer Vision, AutoAttack by Croce et al.[12] is a meta attack combining existing attacks (APGD+FAB) to benefit from the complementarity of search and gradient attacks. The attack has revolutionized the research of robustness assessment in computer vision and led to a significant improvement of novel defense mechanisms. We believe CAA will be as impactful for Tabular Machine Learning as AutoAttack, especially as it was carefully designed to achieve both efficiency for simple cases and effectiveness for challenging cases and has very limited hyper-parameters.
>
> We strongly argue that both attacks required significant design, engineering, and experimentation to find the optimal mechanisms and attacks to combine, and we have demonstrated that our techniques represent a significant leap forward for the community.
>
> **W2 - The implementation of CAA might be complex and resource-intensive, which could limit its practical applicability in some settings:**
>
> Thank you for raising this critical aspect of robustness evaluation. The implementation of CAA brings marginal overhead in terms of implementation, given that the constraint evaluation runs on CPU and is parallelized.
>
> In addition, we have carefully evaluated the impact of CAA in terms of runtime. Compared to the closest attack in terms of performance, MOEVA, CAA is significantly cheaper and faster to run. In Table 3, we showed that CAA is up to 5 times faster than MOEVA, and is always less expensive to run than MOEVA on 4 datasets. The only case where CAA is marginally more resource intensive is on CTU, but the overhead is between 7.76% and 13.74%, respectively for VIME and TabNet architectures.
>
> Thank you for opening this discussion, we will incorporate it in the final manuscript within a limitation section (cf common authors's response, C4). We discuss there the cases where CAA could be more expensive, and will suggest good practices for practitioners to use CAPGD and CAA to their fullest potential.
>
> **W3 - Lack of evaluation against defense mechanisms: There have been many works proposed since Madry Adversarial Training. It is not fair to say that it is the only reliable defense against evasion attacks:**
>
> Thank you for raising this critical point. There may have been a misunderstanding as we do not claim that Madry's Adversarial Training (AT) is the only reliable one, but that AT with all its improvements is. Since then, the robustbench benchmark [A] has continuously updated its leaderboard with new defenses, but all the effective ones are based on adversarial training and some data augmentation mechanisms. Some have proposed AT + Cutmix [B], others AT + Generative models (for example with 20M synthetic data [C]).
> To validate the effectiveness of CAA on stronger defenses we have implemented 5 new synthetic data for Tabular ML in combination with Adversarial Training: AT + TVAE [D], AT + WGAN [E], AT + TableGAN [F], AT + CT-GAN [D] and AT + GOGGLE [G]. We evaluated CAA on these 5 defenses and report in Table 4 of the authors' response PDF the best robustness achieved by these defenses, compared to the robustness by the vanilla Madry AT on all our datasets and models. The results are the average over 5 seeds run to ensure reliable evaluation.
>
> Our new extensive experiments show that these new defenses can significantly improve the robustness of the models to CAA, but that our new attack remains effective for URL and LCLD datasets across all architectures, and for WIDS on TabTransformer and STG architectures.
>
> Thank you for suggesting this study, we will discuss these new results in the appendix of the final version of the manuscript.
>
> [A] Croce et al. "Robustbench: a standardized adversarial robustness benchmark."(2020).
>
> [B] Yun et al. "Cutmix: Regularization strategy to train strong classifiers with localizable features." ICCV, (2019).
>
> [C] Wang et al. "Better diffusion models further improve adversarial training." ICML. PMLR, (2023).
>
> [D] Xu et al. Modeling tabular data using conditional GAN. NeurIPS, (2019).
>
> [E] Arjovsky et al. Wasserstein GAN. CoRR, abs/1701.07875, (2017)
>
> [F] Park et. al. Data synthesis based on generative adversarial networks. VLDB Endowment, (2018)
>
> [G] Liu et al. GOGGLE: Generative modelling for tabular data by learning relational structure. ICLR, (2022)

---

> > ### Author Response · Authors · 2024-08-14
> > **Summary of our improvements**
> >
> > Dear reviewer,
> > we thank you again for your constructive feedbacks, and your suggestions to improve our work.
> >
> > We would like to highlight that we have addressed during this rebuttal all your concerns and interrogations and we have provided additional insights and results to support our claims.
> > In particular, we have adressed the weaknesses you raised as follows:
> >
> > *  W1: Lack of novelty and contribution.
> >
> > => A1: We have elabotated on the complex process of desiging our new CAPGD attack, that required analyzing multiple improvement mechanisms, their interactions, thus leading to only including in CAPGD the most relevant mechanisms. Our CAPGD design is the best that could be achieved for gradient attacks in tabular ML. We also detailed the inception of our new meta-attack, CAA and explained why its design is not straightforward and required the analysis, selection and evaluation of 10 existing attacks.
> >
> > * W2: The implementation of CAA might be complex and resource-intensive
> >
> > => A2: We elaborated on our previous analysis on the computation cost and execution time of CAA compared to MOEVA (Table 3 of our manuscript) and explained that CAA is marginally more ressource-intensive than MOEVA in one over 4 datasets and significantly more efficient than MOEVA in the 3 remaining datasets.
> >
> > * W3: Lack of evaluation against defense mechanisms
> >
> > => A3: We have implemented 5 new defenses using extensive data augmentation and adversarial training. Our new defenses leverage some of the best and most recent generative models of tabular data and required the training of complex generative models to generate 100 times more synthetic examples and achieve the best adversarial training defenses. Our new extensive experiments show that these new defenses can significantly improve the robustness of the models to CAA, but that our new CAA attack remains effective for URL and LCLD datasets across all architectures, and for WIDS on TabTransformer and STG architectures.
> >
> > If you find our answers and discussion satisfactory, we would greatly appreciate it if you could increase your score accordingly. If there are any remaining issues or questions, we would be more than happy to address them before the discussion period ends.
> >
> > Thank you again for your insights and the discussion points you raised.

---

### Official Review · Reviewer_XFcC · 2024-07-11

**Soundness:** 3
**Presentation:** 3
**Contribution:** 3
**Rating:** 7
**Confidence:** 3

**Summary:**

This paper considers the evaluation of the robustness of deep learning models applied to tabular data. The authors introduce two novel adversarial attack methods: Constrained Adaptive Projected Gradient Descent (CAPGD) and Constrained Adaptive Attack (CAA). These methods are designed to exploit the vulnerabilities of tabular data models, which often include categorical features, immutability constraints, and feature relationship constraints that are not typically considered in attacks designed for computer vision (CV) or natural language processing (NLP).

CAPGD improves on existing gradient-based attacks by incorporating adaptive mechanisms and eliminating the need for parameter tuning, significantly enhancing the success rate and efficiency of generating valid adversarial examples. CAA further combines CAPGD with the Multi-Objective Evolutionary Adversarial Attack (MOEVA), optimizing both effectiveness and computational cost. The paper demonstrates the superior performance of CAA across various datasets and models, suggesting it should become a standard test for evaluating the robustness of tabular models.

**Strengths:**

- The paper is well-motivated and logically structured, contributing significantly to advancing adversarial machine learning, particularly in the domain of tabular data.
- The proposed CAA method is SOTA on effectiveness and efficiency, making it a good benchmark for testing the robustness of tabular data models, similar to the role of AutoPGD/AutoAttack in computer vision tasks.
- The paper includes extensive and systematic experiments, providing solid empirical evidence for the effectiveness of the proposed attacks.

**Weaknesses:**

- The paper does not provide a clear explanation of the penalty function in Equation (4). A more detailed clarification is needed to understand how this function is formulated and applied.
- The description of constraints in Equations (1) and (2) is somewhat abstract and difficult to interpret. The authors should refine the descriptions of these constraints to improve clarity.

**Questions:**

- Are there any other constraints in tabular data besides categorical features, immutability, and feature relationship constraints?
- Why are gradient attacks ineffective on the CTU dataset?
- Why is feature engineering necessary for the datasets? Have all four datasets considered in the paper undergone feature engineering? It seems that the WiDS dataset has not undergone feature engineering. Is that correct?

---

> ### Author Rebuttal · Authors · 2024-08-07
>
> Thank you for your positive feedback and your praise for the extensiveness and significance of our work in advancing tabular adversarial machine learning. We appreciate your interest and would be happy to provide further explanations to address any questions you have.
>
> **W1 - Clarification of the penalty function:**
>
> The penalty function transforms each constraint formulation into a differentiable loss function to be minimized by gradient descent. Let's consider one complex constraint from the LCLD credit scoring use case: *The term of the loan can only be 36 or 60 months and the number of open accounts is lower than the number of allowed accounts for this client.*
>
> Such a constraint can be formally written as (term ∈ {36, 60}) ∧ (open_acc ≤ total_acc). The AND operator **∧** is equivalent to a sum of loss, while the element of set operator c∈{a,b,...} is equivalent to multiple OR operators, that are described as min(|c-a|,|c-b|,...) in a loss function. Finally, the a≤b operator is equivalent to a min(0,a-b) in a loss function.
>
> Hence, the complex constraint translates as the following penalty:  min(|term − 36|,|term − 60|)+ max(0, open_acc − total_acc)
>
> Thank you for requesting these clarifications. We will update the final version accordingly and provide a meaningful example for each use case.
>
> **W2 - Clarification of the Equations (1) and (2):**
>
> The grammar in equations (1) and (2) are inspired by the work of Simonetto et al. [33] where they demonstrate the completeness of this grammar and its ability to cover all linear constraints.
>
> To elaborate on the two equations, equation (1) means that a constraint formula ω can either be an intersection (∧), or a union (∨) of two other constraint formulae ω1, ω2, or ω can be a comparison operator between two values ψ, or ω can be the feature $f$ equals a value of the set {ψ1 ... ψk}.
>
> Then equation (2) details the numeric expressions that are supported by the grammar.
> A numeric expression ψ can be constant, an operation between two other numerical expressions ψ1 and ψ2, or a specific feature f, or the value for f of the clean sample $x_i$.
> The difference between $f_i$ and $x_i$ is that $f_i$ corresponds to the current value of the evaluated example and $x_i$ corresponds to its original value in the clean example.
> This seemingly simple grammar allows very large recursive combinations and covers all the relations found in the features of our datasets.
>
> In this grammar, the symbol $\in$ represents a type of constraint, and not that $f$ is a value.
> The constraint $f \in \{ψ_1, ..., ψ_k\}$ is equivalent to $(f=ψ_1) \lor (...) \lor (f=ψ_k)$
> Hence we can simplify the grammar as follows:
>
> $\omega := \omega_1 \land \omega_2 \mid \omega_1 \lor \omega_2 \mid \psi_1 \succeq \psi_2$ (1)
>
> $\psi := c \mid f \mid \psi_1 \oplus \psi_2 \mid x_i$ (2)
>
> To improve the readability of these equations, we will include the above detailed explanation in the final manuscript, and we have prepared an exhaustive table, Table 1 of the Author Rebuttal PDF, with examples of each type of constraint of the grammar, with real-world examples, and how they are converted into a penalty function. We will also include this table in the updated manuscript.
>
> **Q1 - Are there any other constraints in tabular data besides categorical features, immutability, and feature relationship constraints?**
>
> To the best of our knowledge, the only constraints related to tabular features are the ones we handle: types (including discrete, categorical, and binary), immutability, boundaries (minimum and maximum possible values), and feature relationships.
> Other constraints could be considered, but they are related to the threat model and the capabilities of the attackers, and hence outside the scope of the study. For example, the constraints on the budget of the attacker and the cost of changing a feature (for example an attacker could not change the feature of his current balance, without having sufficient resources to actually update his real account balance, or to change its address without the cost of moving its real address).
>
> **Q2 - Why are gradient attacks ineffective on the CTU dataset?**
>
> Thank you for raising this comment. Indeed, CTU dataset exhibits a large number of constraints compared to the other datasets, and some are particularly challenging. We show in the common answer (C1) that these constraints hinder gradient attacks and are harder to optimize.
>
> **Q3- Why is feature engineering necessary for the datasets? Have all four datasets considered in the paper undergone feature engineering?**
>
> You are right, each dataset has been proposed by different research teams with their own pre-processing and feature engineering. We did not run any additional feature engineering.
>
> However, some datasets rely heavily on raw measurements, for example, the Botnet CTU dataset with features related to a number of connections in ports and traffic load; or WIDS dataset where the features are numerical biological data (eg. albumin_apache about albumin concentration in g/L). Other datasets were designed by their authors with more feature engineering. In the LCLD credit scoring dataset, some features (grade, subgrade, fico_range_low) are scores computed by Lending Club to grade the customer and the loan, and are the result of feature processing and engineering of raw features.
> Therefore, the four use cases cover different levels of feature engineering, with datasets as you pointed out requiring little feature engineering (WIDS), and datasets built with more advanced feature engineering (LCLD).
>
> Thank you for raising this point. We will update the appendix section related to the dataset and provide clarifications and relevant references to the feature engineering process of each dataset.

---

> > ### Author Response · Authors · 2024-08-14
> > **Summary of our improvements**
> >
> > Dear reviewer,
> > we thank you again for your constructive feedbacks, and your support of our work.
> > We would like to highlight that we have addressed during this rebuttal all your concerns and interrogations and we have provided additional insights and results to support our claims.
> > In particular, we have adressed your interrogations as follows:
> >
> > *  Q1: Are there any other constraints in tabular data besides categorical features, immutability, and feature relationship constraints?
> >
> > => A1: To the best of our knowledge, our work covers all the constraints related to tabular data.
> >
> > * Q2: Why are gradient attacks ineffective on the CTU dataset?
> >
> > => A2: We provided a detailed analysis in the common answer with an analysis of the constraints of CTU. CTU is robust because of the number of constraints and the number of features involved in each constraint.
> >
> > * Q3: Why is feature engineering necessary for the datasets?
> >
> > => A3: We did not run any additional feature engineering on the datasets. Each dataset was designed by a different source and may have required dedicated feature engineering. We were not involved in this step.
> >
> > If you find our answers and discussion satisfactory, we would greatly appreciate it if you could increase your score accordingly. If there are any remaining issues or questions, we would be more than happy to address them before the discussion period ends.
> >
> > Thank you again for your insights and the discussion points you raised.

---

> > > ### Comment · Reviewer_XFcC · 2024-08-14
> > >
> > > I thank the authors for the detailed response. My concerns have been well resolved. In addition, I appreciate the additional experimental results reported in the rebuttal, which demonstrate the practicality of the proposed attacks. Therefore, I would like to increase my score from 6 to 7 accordingly.

---

### Author Rebuttal · Authors · 2024-08-07

We thank the reviewers for their comments. The reviewers agree on the importance of the problem we tackle and are satisfied with the comprehensiveness of our study and analyses.

Our work proposes the most effective and efficient attacks for Tabular Machine Learning in constrained domains. Our new attack CAA is up to 5 times faster than the SOTA search attack MOEVA and up to 83% percentage points more effective than the SOTA gradient attack CPGD.

To address the reviewers' feedback, we implemented and evaluated 5 new defenses against CAA, provided a generalization study on 2 new models, and analyzed in detail the constraints of the CTU case. These new results are in the attached PDF.

We address below the common comments of the reviewers:

**C1- Novelty of the work (JEHw, aMTd):**

We would like to clarify that research on adversarial robustness for tabular ML is still in its infancy. Our work investigates how adaptive attacks and meta-attacks can form a new and strong standard for such robustness assessment. In computer vision, comparable endeavors had a significant impact (e.g. AutoAttack [12]) and yielded de-facto evaluation standards (e.g. RobustBench [A]). We aim to push such needed advances for tabular ML, while considering its specificities — i.e. the existence of validity constraints.

However, the development of adaptive mechanisms for tabular attacks is not straightforward: blindly combining existing mechanisms (developed for computer vision) yields suboptimal results. Therefore, our work carefully investigates specific adaptations, including a tabular-specific repair mechanism, to form a novel optimized attack (CAPGD).

Furthermore, the development of a meta-attack requires careful selection of the baseline methods. We are the first to investigate this question, through extensive experimentation and the demonstration that not all attacks are needed (some attacks are subsumed by others). We reveal that combining CAPGD with MOEVA yields the best comprehensiveness-efficiency trade-off. CAA, the resulting meta attack, therefore acts as the new baseline for adversarial attacks on tabular ML models.

**C2- Analysis of CTU constraints: (XFcC, 5k6D)**

CAPGD and MOEVA fail to generate adversarial examples on CTU for 2 out of 5 models. CTU dataset has a large number of constraints compared to the other datasets, and some are particularly challenging. We argue that some of these constraints hinder gradient attacks and are harder to optimize.

To confirm our hypothesis and provide additional insights, we split the constraints of CTU based on their type into 4 buckets (Table 2 of the Author Response PDF).

First, we ran an ablation study, where we ignored one bucket of constraint at a time. Next, we studied the success rate when we considered each bucket separately. Finally, we reported the impact of the number of constraints to optimize from CG3, the largest bucket.

The results in Table 3 show that for gradient attacks, removing one type of constraint is not enough to improve the success rate. Constraints across multiple remaining categories are not satisfied. The individual bucket study confirms that only when considering constraints of type CG2 alone, CAPGD improves its success rate (in VIME and TabNet). When only considering CG3 constraints, reducing the number of constraints improves the success rate (by reducing robust accuracy from 95.3% when considering 100% of CG3 constraints to 84.5% and 43.0% respectively when considering 50% and 10% of the constraints).

We hope this fine-grained constraint analysis addresses the interrogations of reviewers #XFcC and #5k6D, and confirms that for CTU dataset, both the number of constraints and the complexity of individual constraints make the constrained adversarial optimization challenging for gradient attacks.

Thank you for pointing out this pattern, we will extend the appendix with a section dedicated to the aforementioned analysis of the constraints of CTU.

**C3- Generalization of our approach to shallow/tree models: (5k6D, aMTd)**

We train Random Forests (RF) and XGBOOST models to achieve the best performance on our datasets and we present in Table 5 of the attached PDF, the robust accuracy of both models against CAA for the four datasets in two settings: (1) Direct attack where CAA (using its search component MOEVA) attacks directly the RF and XGBOOST models, and (2) Transfer attacks, where we craft the examples on our deep learning (DL) models and evaluate them in the RF and XGBOOST models.

Our new study shows that (1) DL models of our study achieve comparable clean performance to the shallow models, (2) both RF and XGBOOST models are vulnerable to direct CAA attacks (down to 9.1% of robust accuracy on LCLD XGBoost), and (3) CAA attacks on DNN transfer to RF (down to 5.3% robust accuracy) and XGBoost (down to 9.4% robust accuracy) models.

This study confirms the relevance and significance of our attacks on tabular models, including undifferentiable models.

**C4- Dedicated "Limitations" section: (All)**

We have scattered the limitations of our approaches across the paper and we summarize them for completeness.

- *Marginal overhead of CAA:* In settings where CAPGD fails to attack tabular models, CAA can exhibit a computation overhead (<14%) compared to MOEVA. However, in 4/5 evaluated settings, CAA is faster than MOEVA (up to 5 times).
- *CAPGD effectiveness with complex constraints:* CAPGD effectiveness drops when increasing the constraint's complexity such as the number of constraints or the number of features involved in each constraint.
- *Coherence of constraints:* The mechanisms of CAA assume that the constraints definitions are sound. Incoherences between boundary constraints and feature relation constraints can lead to invalid adversarial examples with large EPS budgets.

We will introduce both the study of tree-based models and the dedicated limitation section in the final version of the paper.

---

### Decision · Program_Chairs · 2024-09-25

**Decision:**

Accept (spotlight)

**Comment:**

**Summary of the paper**

This paper proposes a new attack method for generating adversarial perturbations against deep neural networks for tabular data. Specifically, the authors propose adaptive mechanisms that overcome the problems of existing gradient-based attack methods. Then, they strengthen their attack method by combining it with an existing search-based attack. With a set of experiments, the authors demonstrate the effectiveness of the attack.

**Summary of the discussion**
* The reviewers observed several strengths of this paper, including its novelty, the importance of the problem domain, and the extensive number of experiments.
* Reviewer XFcC pointed out two weaknesses.
  * First, the definition of the penalty function needs to be clarified.
  * Second, some descriptions are difficult to understand, and more clarification is needed.
  * The authors provided more detailed explanations in the rebuttal, saying they would address the reviewer’s concern in the revised version.
  * After the discussion, the reviewer increased the rating from 6 to 7.
* Reviewer JEHw pointed out three weaknesses.
  * First, the reviewer believes the proposed method lacks novelty. Regarding the novelty of the work, the authors emphasized that finding the appropriate attack method for tabular data was already the first challenge they overcame and that combining existing attacks is an active line of research.
  * Second, the proposed method might be too complicated and resource-intensive. Regarding resource consumption, the authors mentioned that the proposed method only induces marginal overhead if the constraint evaluation is parallelized.
  * Third, the reviewer says the authors only compared to old defense mechanisms. For the last weakness, the authors admitted that it is a critical point and then provided more results against five additional defenses in the revised version.
  * After the discussion, the reviewer increased the rating from 4 to 6.
* Reviewer 5k6D pointed out five weaknesses.
  * First, the algorithm description is not clear. Regarding the first weakness, the authors clarified the part the reviewer asked for, i.e., the role of the repair operator.
  * Second, the analysis of the result can be improved further. Regarding the second weakness, the authors admitted the reviewer's concerns and provided some rough analysis of the part the reviewer mentioned.
  * Third, there is no dedicated section for the proposed method's limitations. Regarding the third concern, the authors added a dedicated section for the limitation of their work.
  * Fourth, there are typos and inconsistencies in notations. Regarding the minor issues in writing, the authors addressed the concerns by fixing typos.
  * Last, the proposed method's impact is limited to a specific domain. Regarding the reviewer's point that tree-based models outperform deep neural networks on tabular data, the authors claimed that the proposed method has potential application to any existing model, including tree-based models.
  * After the discussion, the reviewer increased the rating from 6 to 7.
* Reviewer aMTd pointed out three weaknesses of the paper.
  * First, the proposed idea has already been explored by existing work. Regarding the first weakness, the authors admitted that the idea of a differentiable penalty function is not their novel contribution. However, they also point out that the penalty function alone is not enough for an effective adversarial attack, emphasizing their contribution in improving the mechanism.
  * Second, the proposed method looks like a mere improvement of an existing algorithm. Regarding the second concern, the authors list four novel mechanisms they introduced in their method. The authors also pointed out that finding the proper attack method was already a big challenge when attacking tabular data.
  * Third, the reviewer raised a concern about the authors’ meta-attack design. Regarding the last concern, the author appreciated the reviewer’s concern regarding their meta-attack design and emphasized its contributions.
  * After the discussion, the reviewer increased the rating from 4 to 5.

**Justification of my evaluation**

* My rating is 6 (or 7 with lower confidence).
* To the best of my knowledge, adversarial attacks on tabular data have not been explored enough, and this work contains enough contributions for acceptance.
* During the discussion, the authors communicated very effectively with the reviewers. All the reviewers increased their ratings after the discussion, which is impressive.

I recommend accepting this work (spotlight). However, I’m unsure whether I can recommend it for the spotlight. I’d ask the SAC for some guidance.